



# Development of adjoint-based ocean state estimation for the Amundsen and Bellingshausen Seas and ice shelf cavities using MITgcm/ECCO

Yoshihiro Nakayama[1,2], Dimitris Menemenlis[1], Ou Wang[1], Hong Zhang[1], Ian Fenty[1], and An T. Nguyen[3]

[1]Jet Propulsion Laboratory, California Institute of Technology, Pasadena, California, USA.
[2]Institute of Low Temperature Science, Hokkaido University, Sapporo, Hokkaido, Japan.
[3]The University of Texas at Austin, Austin, Texas, USA

**Correspondence:** Yoshihiro Nakayama (Yoshihiro.Nakayama@lowtem.hokudai.ac.jp)

**Abstract.** The Antarctic coastal ocean is impacting sea level rise, deep-ocean circulation, marine ecosystems, and global carbon cycle. To better describe and understand these processes and their variability, it is necessary to combine the sparse available observations with best-possible numerical descriptions of ocean circulation. In particular, high ice-shelf melting rates in the Amundsen Sea have attracted many observational campaigns and we now have some limited oceanographic data that capture

seasonal and interannual variability during the past decade. One method to combine observations with numerical models that can maximize the information extracted from the sparse observations is the adjoint method, as developed and implemented for global ocean state estimation by the Estimating the Circulation and Climate of the Ocean (ECCO) project. Here, for the first time, we apply the adjoint-model estimation method to a regional configuration of the Amundsen and Bellingshausen Seas, Antarctica, including explicit representation of sub-ice shelf cavities. We utilize observations available during 2010–2014,

including ship-based and seal-tagged CTD measurements, moorings, and satellite sea-ice concentration estimates. After 20 iterations of the adjoint-method minimization algorithm, the cost function, here defined as a sum of weighted model-data difference, is reduced by 65% relative to the baseline simulation by adjusting initial conditions, atmospheric forcing, and vertical diffusivity. The sea-ice and ocean components of the cost function are reduced by 59% and 70%, respectively. Major improvements include better representations of (1) Winter Water (WW) characteristics and (2) intrusions of modified Circumpolar

Deep Water (mCDW) towards the Pine Island Glacier. Sensitivity experiments show that ∼40% and ∼10% of improvements in sea ice and ocean state, respectively, can be attributed to the adjustment of air temperature and wind. This study is a preliminary demonstration of adjoint-method optimization with explicit representation of ice-shelf cavity circulation. Despite the 65% cost reduction, substantial model-data discrepancies remain, in particular with annual and interannual variability observed by moorings in front of the Pine Island Ice Shelf. We list a series of possible causes for these residuals, including limitations of

the model, the optimization methodology, and observational sampling. In particular, we hypothesize that that residuals could be further reduced if the model could more accurately represent sea ice concentration and coastal polynyas.





## 1 Introduction

The ice shelves and glaciers in the Amundsen Sea (AS) and Bellingshausen Sea (BS) are melting and thinning rapidly with consequences for global sea level rise and changes in ocean circulation and global carbon cycle (e.g., Arrigo et al., 2008; Pritchard et al., 2012; Paolo et al., 2015; Bronselaer et al., 2018; Rignot et al., 2019). Basal melting of these ice shelves is caused by warm modified Circumpolar Deep Water (mCDW, 0.5–1.5°C), which intrudes onto the continental shelf toward ice shelf cavities following submarine glacial troughs (Fig. 1) (e.g., Jacobs et al., 1996; Walker et al., 2007; Jacobs et al., 2011; Nakayama et al., 2013; Walker et al., 2013; Dutrieux et al., 2014). For this reason, multiple oceanographic observational campaigns have been collected by the international community to understand the mechanism of mCDW intrusions onto the AS continental shelf and towards ice shelf cavities. As part of these efforts, we now have some limited oceanographic data that capture seasonal and interannual variability during the past decade (e.g., Jacobs et al., 2011; Nakayama et al., 2013; Dutrieux et al., 2014; Heywood et al., 2016; Kim et al., 2017; Webber et al., 2017; Mallett et al., 2018).

Recent observations as well as modeling studies reveal that mCDW pathways, ice shelf-ocean interaction, the thermocline depth, and ocean bathymetry below Pine Island Ice Shelf (PIIS) are important for controlling the PIIS melt rate (e.g., Schodlok et al., 2012; Dutrieux et al., 2014; De Rydt et al., 2014; St-Laurent et al., 2015; Dinniman et al., 2016; Jourdain et al., 2017; Kimura et al., 2017; Webber et al., 2019). The thermocline depth was ∼200 m deeper in 2012 compared to other years (e.g., 1994, 2007, 2009, and 2010, see Fig. 2A in Dutrieux et al. (2014)), which reduced the PIIS melt by ∼ 50%. After 2012, the thermocline shoaled by 200m returning to its more commonly observed depth of ∼350 m (Webber et al., 2017). It is suggested that this thermocline variability was caused by changes in local and remote surface wind and buoyancy forcing (Dutrieux et al., 2014; Webber et al., 2017).

To better describe and understand these processes and their variability, it is necessary to combine the sparse available observations with best-possible numerical descriptions of ocean circulation. One method to combine observations with numerical models that can maximize the information extracted from the sparse observations is the adjoint method, as developed and implemented for global ocean state estimation by the Estimating the Circulation and Climate of the Ocean (ECCO) project. To date, the ECCO project has produced ocean state estimates based on Circum-Antarctic or global model configurations (e.g., Mazloff et al., 2010; Forget et al., 2015; Zhang et al., 2018; Fukumori et al., 2020). Employing the adjoint model produced by automatic differentiation (Giering and Kaminski, 1998) and utilizing temporally-varying oceanographic observations, these ocean state estimates are capable of simulating the large-scale evolution of the Southern Ocean consistent with the available observations. Many observational and modeling studies have been conducted to understand Southern Ocean gyre dynamics, subsurface ocean circulation, the southern shift of various fronts around Antarctica, etc. (e.g., Gille et al., 2016; Jones et al., 2016; Tamsitt et al., 2017; Nakayama et al., 2018; Roach and Speer, 2019; Jones et al., 2020). However, despite the importance of Antarctic coastal regions for global climate, existing models fail to accurately reproduce the sparse available observations, likely owing to the difficulty in simulating Antarctic continental shelf regions and sub-ice-shelf-cavity processes (Mazloff et al., 2010; Timmermann et al., 2012; Kusahara and Hasumi, 2013; Nakayama et al., 2014; Rodriguez et al., 2016; Nakayama et al., 2017; Kusahara, 2020).



For other regions of the globe, ocean state estimates based on regional configurations have been successfully developed during the past decades, achieving good model-data agreement and leading to understanding of reginal processes (Fenty and Heimbach, 2013b,c; Verdy et al., 2014; Rudnick et al., 2015; Nguyen et al., 2020; Verdy et al., 2017; Vinogradova et al., 2014). For Antarctic coastal regions, however, the only previous attempt to constrain a model with observations was the study of Nakayama et al. (2017), which used a low-dimensional estimation approach based on the computation of model Green's functions. Here we aim to extend the study of Nakayama et al. (2017) by employing the adjoint method, which permits a larger number of higher dimension control variables than the Green's Function approach. The objective is to obtain a closer fit to the available observations than what was achieved in Nakayama et al. (2017). This objective is challenging due to several difficulties including (1) polar specific processes (ice shelf and sea ice) are highly nonlinear and (2) observational data is limited. The groundwork for making adjoint-method optimization possible in the presence of ice shelf cavities was laid out in the study of Heimbach and Losch (2012), who obtained adjoint sensitivities of sub-ice shelf melt rates to ocean circulation under Pine Island Ice Shelf, West Antarctica.

In this study, we present our attempt at the development of Amundsen-Bellingshausen Seas ocean state estimates (Fig. 2) by employing the adjoint-model-based data assimilation method developed by ECCO for regional and global ocean state estimation (Mazloff et al., 2010; Forget et al., 2015; Zhang et al., 2018; Fukumori et al., 2020). We focus on the years 2010–2014 when oceanographic observations were collected frequently and the largest interannual variability has been observed (Dutrieux et al., 2014; Webber et al., 2017). Our simulations are carried out for a subregion of the global Latitude-Longitude-Polar-Cap (LLC) 270 configuration (Zhang et al., 2018). As the LLC270 horizontal and vertical resolutions are insufficient to resolve critical ocean-ice shelf interaction processes, e.g., eddy transport, mean flow topography interaction, this study can be regarded as a step toward improved parameterization and adjustment of these processes in future global ECCO optimizations.

## 2 Data and methods

### 2.1 Observations

In the Amundsen Sea, oceanographic observational campaigns were carried out in 2010, 2012, and 2014 (Nakayama et al., 2013; Dutrieux et al., 2014; Heywood et al., 2016; Kim et al., 2017). Several mooring observations were also obtained, with the moorings at the PIIS front capturing the largest interannual variability observed in the region between 2009–2014 (Dutrieux et al., 2014; Webber et al., 2017). We also utilize seal-tagged CTD observations obtained in 2014, which contain over 10,000 profiles between February and November (Heywood et al., 2016). In the central part of the Bellingshausen Sea, no oceanographic observations were collected between 2010–2014. Despite that seal-tagged CTD observations are available (Roquet et al., 2013; Zhang et al., 2016), these datasets are not used in the current version of state optimization. For the Antarctic peninsula region, oceanographic observations were collected by the Palmer Antarctic Long-Term Ecological Research project (PAL-LTER, Ducklow et al. (2012)). For sea ice, we use satellite-based estimates of daily sea ice concentration with grid resolutions of 25 km (Cavalieri et al., 1996). The datasets used in this study are summarized in Figs. 2-3 and Table 1.





## 2.2 Numerical model

We employ the Massachusetts Institute of Technology general circulation model (MITgcm), which includes dynamic/thermodynamic
sea-ice (Losch et al., 2010) and thermodynamic ice shelf (Losch, 2008) capabilities. Following the model configuration from
Nakayama et al. (2017), we extract the regional grid from a global LLC270 configuration for the AS and BS regions (Fig. 1).
South of 70°S, the LLC270 configuration uses a bipolar grid; in the AS and BS domain, the horizontal grid spacing is approximately 10 km (Fig. 1). The vertical discretization of the ECCO LLC270 configuration comprises 50 levels varying in
thickness from 10 m near the surface, 70–90 m in the 500–1000-m depth range, and 450 m at the deepest level of 6000 m.
Model bathymetry is derived from the International Bathymetric Chart of the Southern Ocean (IBCSO; Arndt et al. (2013))
and the model ice draft is based on Antarctic Bedrock Mapping (BEDMAP-2; Fretwell et al. (2013)). Following Nakayama
et al. (2017), we simulate the effect of the ice barrier (shown in white indicated by the red arrow in Fig. 1) by limiting sea-ice
transport between the eastern and central AS.

The first guess of the model initial state is a simulated 2010 oceanographic condition based on Green's function approach
(Nakayama et al., 2017). Lateral boundary conditions for hydrography, currents, and sea ice are provided by the ongoing ECCO
LLC270 optimization. The initial guess of surface forcing for 2010–2014 period is provided by ERA-Interim (Dee et al., 2011).
There is no additional freshwater runoff above and beyond the meltwater computed by the MITgcm ice shelf package. The
model parameters used for this state estimate are shown in Table 2.

The MITgcm 4D-Var assimilation system iteratively minimizes the squared sum of weighted model data misfits and control
adjustments by using the adjoint model to adjust the control variables (Wunsch et al., 2009; Wunsch and Heimbach, 2013). The
control vector consists of model initial conditions for temperature and salinity, vertical diffusivity, and all atmospheric states
(air temperature, specific humidity, precipitation, shortwave radiation, longwave radiation, and eastward and northward winds).
The gradient of the cost function is obtained by integrating the adjoint of the tangent linear model backward in time (Le Dimet
and Talagrand, 1986) and is used with the quasi-Newton M1QN3 conjugate-gradient algorithm (Gilbert and Lemaréchal, 1989)
to adjust the control variables to iteratively reduce the cost function toward its minimum. Sea ice concentration is constrained
using a pseudo-sea ice adjoint, whose details are provided in Forget et al. (2015). We also note that background horizontal
viscosity has to be artificially increased at the early stage of the optimization for model stability and we manually lowered the
values of viscosity at iterations 10, 15, and 20 (Tables 2 and 3).

For the static ice shelf component, the freezing/melting process in the sub-ice-shelf cavity is parameterized by the three-
equation thermodynamics of Hellmer and Olbers (1989); Jenkins (1991). We use constant turbulent heat and salt exchange
coefficients for individual ice shelves, which are already adjusted in Nakayama et al. (2017). However, only for Pine Island
and Thwaites, we further modify these coefficients for simulations after iteration 11 (Table 3), as ice shelf melt rates of Pine
Island and Thwaites become too large. Changes of these coefficients do not highly alter on-shelf circulation (see Fig. S18 in
Nakayama et al. (2018)) and adjustments of these coefficients in addition to optimizations based on adjoint sensitivities is
possible.





## 3 Results

### 3.1 Unoptimized simulation (iteration 0)

As we initialize the unoptimized simulation (iteration 0) with simulated oceanographic conditions based on Green's function approach (Nakayama et al., 2017), its 2010 simulated vertical sections show a good agreement with observations (Fig. 4).

Simulated vertical sections present mCDW below 400–500 m and WW above 250–400 m consistent with observations (Fig. 2 in Jacobs et al. (2011) and Fig. 4 in Nakayama et al. (2013)). Similar to Nakayama et al. (2017), slight differences can still be found for WW properties close to the surface (salinity being still too saline (∼0.1 psu)) and PIIS front mCDW properties (∼0.1 psu for salinity and ∼0.2°C for potential temperature, Fig. 4).

We find, however, that the time evolution of iteration 0 between 2010–2014 does not agree well with observations. For exam-

130 ple, oceanographic conditions at the PIIS front in iteration 0 simulation becomes too cold and fresh by ∼2°C and ∼0.25 psu, respectively (Figs. 4 and 5). Furthermore, the horizontal section of potential temperature at 552 m depth illustrates the formation of cold and fresh water masses (∼ −1℃ and 34.4 psu) in the vicinity of the PIIS (the red arrows in Fig. 6). This water spreads along the coast and induces unrealistic cooling in the large area of the AS (Figs. 6 and 7). Simulated time series of potential temperature and salinity at the PIIS front mooring (Fig. 8) shows that these changes occur as a result of intense cooling by the

135 atmosphere in the austral winter of 2013 and this cooling leads to the reduction of the PIIS melt rate by ∼100 Gt yr$^{-1}$ (Fig. 9a).

### 3.2 Model–Observation Differences and improvements

As a result of the iterative optimization, we are able to reduce the cost, which is defined as a sum of weighted model-data difference, by 65% by adjusting initial ocean temperature and salinity, atmospheric surface parameters, and vertical diffusivity (Fig. 2). The cost reduction occurs quicker in the first 10 iterations (Fig. 2). Throughout the optimization, sea ice and ocean

costs are reduced by 59% and 70%, respectively.

#### 3.2.1 Sea ice

In the iteration 20 simulation, spatial patterns of the sea ice concentrations show better agreement with observations. For September, simulated sea ice concentration is overestimated in iteration 0 at the northern model boundary but becomes much closer to observations in iteration 20 (Fig. 10). For March, simulated sea ice area is larger by 0.12 million km$^2$, and 0.13 million

145 km$^2$ in iteration 0 and 0.08 million km$^2$ and 0.05 million km$^2$ in iteration 20 in the AS and BS, respectively (Fig. 10).

#### 3.2.2 Ocean

For the AS, there are two major improvements for the oceanographic condition: (1) representation of mCDW intrusions towards the ice shelf cavities and (2) properties of WW. For the BS, we do not include enough observational data in the current version of the ocean state estimate and are not able to judge the capability of our state estimation.





As model-data difference becomes larger towards the end of the 2010–2014 simulation for unoptimized simulation, we
compare 2014 oceanographic conditions between iterations 0 and 20 simulations to assess improvements. At greater depths,
mCDW penetrates along the submarine glacial troughs towards the Pine Island, Thwaites ice shelf cavities (the red arrows in
Figs.6a, b) in the iteration 20 simulation, qualitatively similar to observations and other model studies (Jacobs et al., 2011;
Nakayama et al., 2013; Dutrieux et al., 2014; Nakayama et al., 2018, 2019). The 552-m potential temperature and salinity
difference between iteration 0 and iteration 20 simulations are ∼0.5°C and ∼0.1 psu along the coast of the AS, respectively
(Fig. 6). Simulated time series of potential temperature and salinity at the PIIS front mooring also show the continuous intrusion
of mCDW into the PIIS cavity from 2010–2014, consistent with observations (e.g., Jacobs et al., 2011; Dutrieux et al., 2014;
Kim et al., 2017; Jenkins et al., 2018; Assmann et al., 2019).

  At shallower depths, the thermocline is located deeper by ∼150 m in the AS in 2014 than in 2010 (Figs.4 and 5), which
seems to be consistent with observations (Jacobs et al., 2011; Nakayama et al., 2013; Dutrieux et al., 2014; Heywood et al.,
2016). The 222-m salinity difference between iteration 0 and iteration 20 simulations shows freshening by 0.05-0.1 mostly
everywhere in the AS (the red arrows in Figs. 7d and e). This is a good improvement as WW tends to become too saline in
most numerical models (e.g., St-Laurent et al., 2015; Nakayama et al., 2018) and good representations of surface hydrographic
conditions as well as stratification are necessary for the better representation of interannual variabilities.

### 3.2.3 Ice shelf

Heat and salt transfer coefficients are kept constant and we do not allow them to change over time. Thus, time series of ice shelf
melt rates simply reflect changes of oceanographic conditions in the ice shelf cavities. We note, however, that these coefficients
are adjusted once at iteration 11 for Pine Island and Thwaites ice shelves (Tables 3 and 4). In iteration 0, simulated time series
of Pine Island and Thwaites ice shelf melt rates show a reduction of ∼100 Gt yr$^{-1}$ between 2010–2014 (Fig. 9). In the iteration
20 simulation, however, both time series of Pine Island and Thwaites ice shelf melt rates become rather stable at ∼110 Gt
yr$^{-1}$ but show slight decreasing trends of ∼4 Gt yr$^{-2}$ and ∼3 Gt yr$^{-2}$, respectively. Simulated Pine Island melt rate shows a
reduction in 2012 by ∼30 Gt yr$^{-1}$ and simulated Thwaites melt rate shows reductions of ∼20 Gt yr$^{-1}$ and ∼50 Gt yr$^{-1}$ in
2012 and 2013, respectively. For ice shelves in the BS, melt rates remain almost constant (e.g., Fig .9).

  Based on observations, it is suggested that melt rates of Pine Island and Thwaites ice shelves should have decreased between
175 2012–2014 due to deepened thermocline (Dutrieux et al., 2014; Webber et al., 2017) and estimated ice shelf melt rates from
oceanographic observations are ∼75 Gt yr$^{-1}$ and ∼40 Gt yr$^{-1}$ based on 2009/2010 and 2012/2014 observations, respectively.
However, these estimates rely on single snapshots of ice shelf front oceanographic observations. Satellite-based estimates of
ice shelf melt rates are 101 Gt yr$^{-1}$ and 98 Gt yr$^{-1}$ for Pine Island and Thwaites ice shelves, respectively. These estimates are
derived from volume flux divergence of Antarctic ice shelves in 2007 and 2008 with 1979 to 2010 surface accumulation and
180 2003 to 2008 thinning (Rignot et al., 2013) and may represent ice shelf melt rates in warm oceanographic conditions in the
eastern AS.

  In general, heat and salt transfer coefficients are already adjusted in Nakayama et al. (2017), and melt rates of ice shelves
in the AS and BS are consistent with satellite-based estimates (Table 4). The interannual variability of the simulated ice shelf





melt rates may be too weakened compared to observations possibly because (1) coarse horizontal and vertical grids used in
this configuration may not allow the realistic representation of ocean cavity circulation and/or (2) observed reduction of ice
shelf melt rates are caused by changes in ice shelf geometry and it can not be simulated in static ice shelf cavity configuration.
Satellite based estimates of time-evolving ice shelf melt rates are required for further comparison.

## 4 Discussion

### 4.1 Sensitivity studies

To investigate the reason for the improvements, we conducted 3 sensitivity experiments (Table 5) as air temperature, precip-
itation, and wind are considered to be main drivers of oceanographic variabilities at the PIIS front region. The total costs are
$2.9 \times 10^6$, $3.1 \times 10^6$, $2.9 \times 10^6$, and $4.1 \times 10^6$ for iteration 20, NoWindAdj, NoPrepAdj, and NoAtempAdj cases, respectively
showing that adjustments of wind and atmospheric temperature play important roles for reducing both sea ice and ocean
costs. Sea ice costs are $1.7 \times 10^6$, $1.6 \times 10^6$, $1.7 \times 10^6$, and $2.9 \times 10^6$ for iteration 20, NoWindAdj, NoPrepAdj, and NoAtempAdj
cases, respectively . Ocean costs are $1.1 \times 10^6$, $1.5 \times 10^6$, $1.1 \times 10^6$, and $1.2 \times 10^6$ for iteration 20, NoWindAdj, NoPrepAdj, and
NoAtempAdj cases, respectively. This implies that adjustment of wind has the strongest impact on the ocean, while adjustment
of air temperature has the strongest impact on sea ice.

Among these sensitivity experiments, the 2014 January mean potential temperature at 552 m depth shows a similar spatial
pattern for all cases for open ocean and in the BS (not shown) and differences can only be found in the AS especially at the
PIIS front (Fig. 11). Spatially averaged 552 m potential temperatures at the PIIS front (averaged for the region enclosed by red
box in Fig 11a) are 0.61°C, 0.23°C, 0.56°C, and 0.53°C for iteration 20 NoWindAdj,, NoPrepAdj, and NoAtempAdj cases,
respectively. Vertically integrated heat contents, which are strongly controlled by thermocline depth (Nakayama et al., 2018),
reduced by 11%, 5%, and 12%, respectively, for NoWindAdj, NoPrepAdj, and NoAtemAdj cases compared to iteration 20
solution. This implies that (1) PIIS front mCDW temperature and thus mCDW pathways as well as strength of intrusions are
dominantly controlled by wind and (2) the PIIS front thermocline depth is influenced rather equally by wind, precipitation, and
air temperature.

### 4.2 Seasonal and interannual variability

Mooring observations at the PIIS front were conducted from 2009–2014, which provide us potential temperature measurements
at various depths (Webber et al., 2017). At depths below 800 m, the observed potential temperature remains rather stable at
~1°C (Fig. 12c). At 600-700 m depths, the potential temperature also remains stable at 1°C and shows gradual cooling and
warming between 2010–2012 and 2013–2015, respectively (Fig. 12). Between 2012–2013, however, potential temperature time
series shows sporadic emergence of cold watermass (~-0.5°C). At 400-500 m depths, the time series of potential temperature
fluctuates between -1.8 °C and 0 °C and seasonal and interannual variabilities are large.





For iteration 0, we find three major differences with respect to the observations (Fig. 12); (1) simulated potential temperature
shows rapid cooling between 2013–2015 and potential temperature at all depths changes from ∼ 1°C to ∼-1 °C, (2) simulated
time series of potential temperature shows sudden emergence of cold water at all depths throughout the simulated period, while
it occurs only for shallower depths for observations, and (3) timing of cooling and warming do not agree with observations.
This sporadic coolings are likely associated with strong wind events which lead to the formation of cold and dense water and
deep convection.

For the iteration-20 simulation, simulated timeseries show improvements at all depths (Fig. 12). At greater depths (800–900
m), the potential temperature remains rather stable at ∼1°C consistent observations (Fig. 12c) and the sudden emergence of
cold water only occurs at shallower depths. However, the long term and short term variabilities have large differences between
observations and the iteration-20 simulation, and the timing of cooling and warming still do not agree with observations.
One possible reason for the difference is deficiencies in the simulated sea ice concentration near the coast. In our current
configuration using a pseudo-sea-ice adjoint sensitivity, we are not able to directly adjust sea ice concentration: simulated sea
ice concentration at the PIIS front remains almost the same between the iteration 0 and iteration-20 simulations (Fig. 12).

## 5 Conclusions

In the previous work, Nakayama et al. (2017) employed a Green's function approach to adjust a numerical simulation of 2010
AS conditions close to observations. However, we find that continuation of the Nakayama et al. (2017) set-up until the year
2014 leads to unrealistic cooling and freshening at the PIIS front and other coastal regions of the AS (Figs. 4–6, 8).

In this work, we develop an Amundsen Bellingshausen Sea ocean simulation following Nakayama et al. (2017) and employ
the ECCO ocean state estimation tools based on adjoint sensitivities (Forget et al., 2015; Zhang et al., 2018) to develop an
ocean state estimate for the AS and BS for the time period of 2010–2014. We choose this time window because the largest
interannual variability was observed after first observations in 1994 (Dutrieux et al., 2014) and a good amount of oceanographic
observations is available. After 20 iterations, cost, which is defined as a sum of weighted model-data difference, is reduced by
65% by adjusting initial condition, atmospheric forcing, and vertical diffusivity (Fig. 2). The iteration 20 simulation can sim-
ulate oceanographic conditions much closer to observations for the 2010–2014 period compared to the unoptimized iteration
0 simulation. The main improvements are (1) simulated sea ice extent for the AS and BS, (2) simulated WW properties and
thermocline depths in the AS, and (3) simulated mCDW intrusions towards AS ice shelf cavities and their pathways (Figs. 5-7).
Despite the improvements listed above, seasonal and interannual variability of oceanographic conditions at the PIIS front is
not simulated well compared to the mooring observations and it remains difficult to simulate seasonal and interannual changes
of oceanographic conditions on the AS continental shelf (Fig. 12).

There are several lines of investigation that can improve upon the technical foundation discussed hereinabove. This includes
new sea-ice adjoint optimization code (Fenty and Heimbach, 2013a; Bigdeli et al., 2020), improved methods of calculating
costs to put more emphasis on the seasonal and interannual variabilities (Forget et al., 2015), adding other oceanographic
datasets not used in the current optimization such as moorings and instrumented pinnipeds (Roquet et al., 2013; Assmann

et al., 2019), and more careful estimation of model and data prior uncertainty. Considering the grid resolution selected for this regional model (10-km horizontal grid spacing), this work is a step towards the improved representation of ice-shelf ocean interaction in the ECCO (Estimating the Circulation and Climate of the Ocean) global ocean retrospective analysis as well as 250  current-generation IPCC (Intergovernmental Panel on Climate Change) global climate models.

*Data availability.*  The model code, input, and results of iteration 20 are available at https://doi.org/10.5281/zenodo.4541036. They are also available at https://ecco.jpl.nasa.gov/drive/files/ECCO2/LLC270/ABS_ADJOINT/results.

*Author contributions.*  Y.N. conceived the study, conducted the ocean modeling, and wrote the initial draft of the paper. D.M., O.W., H.Z., and A.N. contributed to the technical development of regional Amundsen and Bellingshausen optimization. Y.N., D.M., O.W., H.Z., A.N., 255  and I.F. discussed the results and implications and commented on the manuscript at all stages.

*Competing interests.*  The authors declare no competing interests.

*Acknowledgements.*  The research was carried out at the Jet Propulsion Laboratory, California Institute of Technology, under a contract with the National Aeronautics and Space Administration (NASA). Support was provided by an appointment to the NASA Postdoctoral Program; the NASA Cryosphere program; and the NASA Modeling, Analysis, and Prediction program. Computations were carried out at the NASA 260  Advanced Supercomputing facilities. This work was also supported by the fund from Grant in Aids for Scientific Research (19K23447) of the Japanese Ministry of Education, Culture, Sports, Science, and Technology. We thank Karen Heywood, Ben Webber, Stan Jacobs, Tae Wan Kim, Catherine Walker for their support for finding and accessing observational datasets. We also thank Patrick Heimbach, Martin Losch, and Matthew Mazloff for their technical suggestions.



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



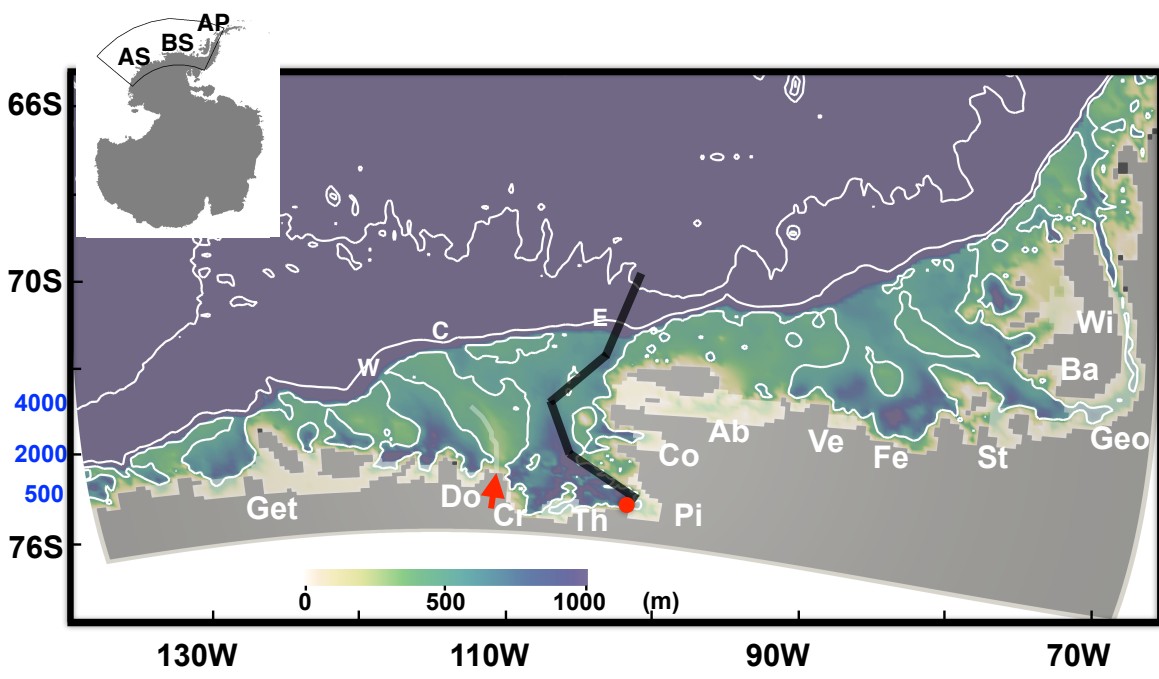

**Figure 1.** Model bathymetry (color) with contours of 500, 2000, and 4000 m in white. The inset (left top) shows Antarctica with the region surrounded by a black line denoting the location of the enlarged portion. AS, BS, and AP denote the Amundsen Sea, Bellingshausen Sea, and Antarctic Peninsula region, respectively. The ice shelves are indicated with transparent white patches and acronyms are summarized in Table 4. Letters E, C, and W denote the submarine glacial troughs located on the eastern AS continental shelf. Transparent white patches (see the red arrow between Do and Cr) indicates the location of grounded icebergs and landfast ice. This white region is treated as a barrier in the sea ice model and we do not allow sea ice exchange crossing this region. The thick black line represents the vertical section shown in Figs. 4 and 5.



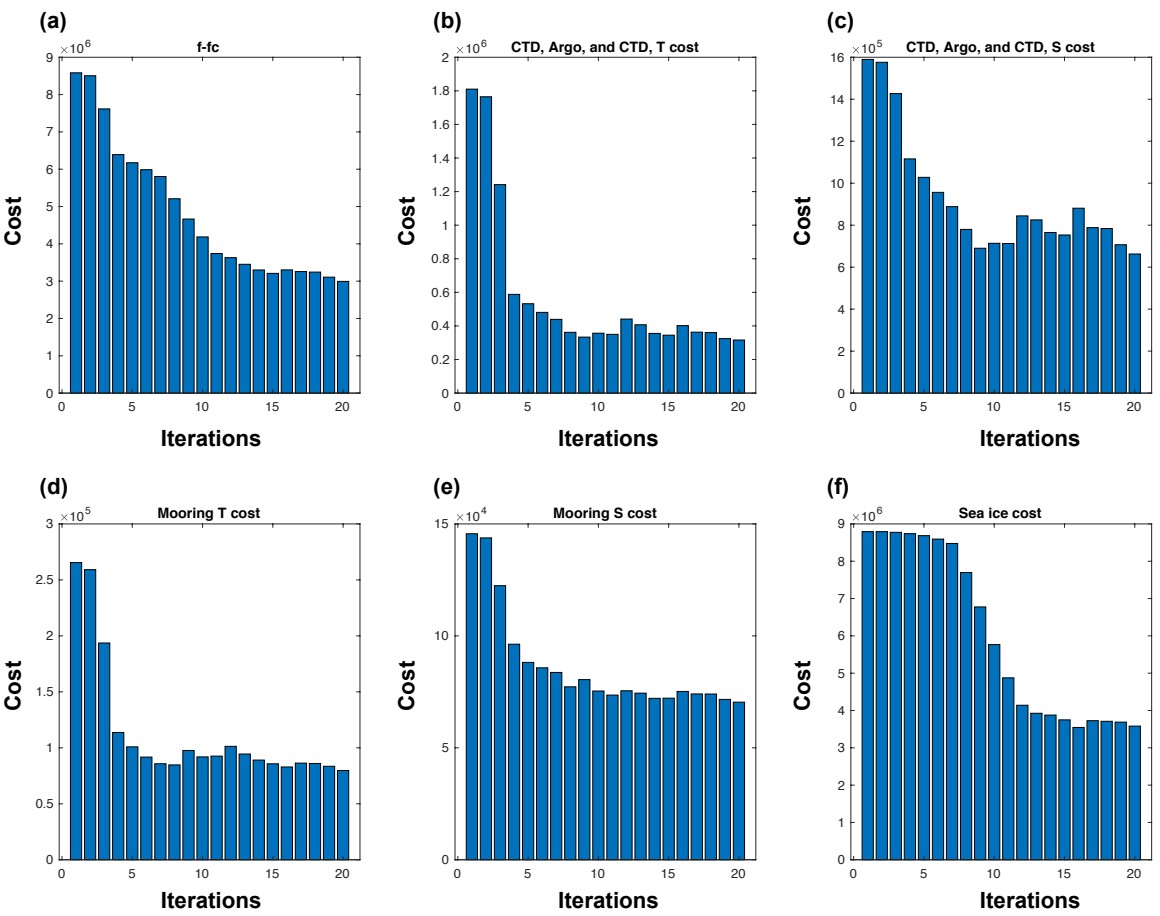

**Figure 2.** Evolution of (a) total, (b) ship-based, Argo, and seal-tagged CTD temperature, (c) ship-based, Argo, and seal-tagged CTD salinity, (d) mooring temperature, (e) mooring salinity, and (f) sea ice costs.

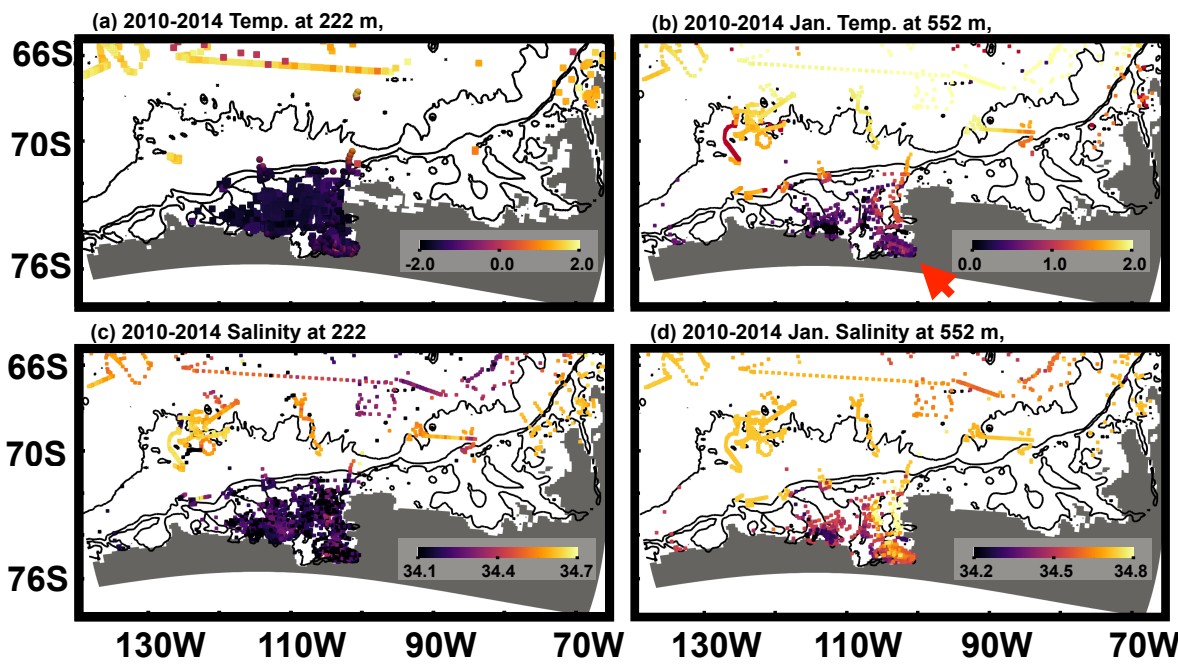

**Figure 3.** (a) 222-m and (b) 552-m potential temperature used for model-data difference calculation and (c) 222-m and (d) 552-m salinity used for model-data difference calculations. Bathymetric contours of 500, 2000, and 4000 m are shown in black. The red arrow indicates the PIIS front region.



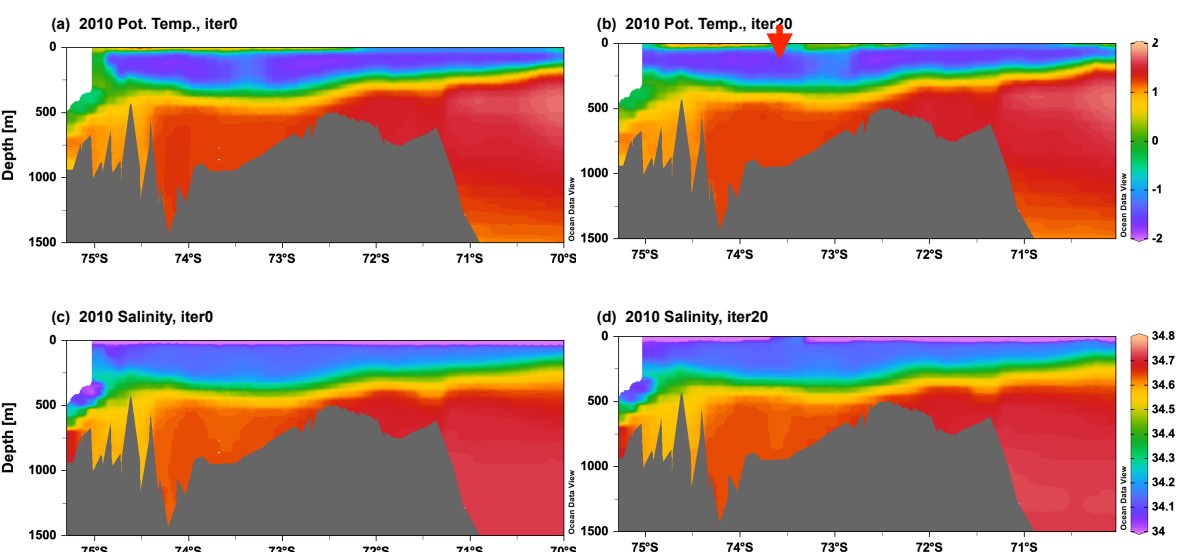

**Figure 4.** Simulated vertical sections of monthly mean potential temperature (top) and salinity (bottom) in January 2010 along the thick black line in Figure 1 for the (left) unoptimized and (right) iteration 20 simulations. The red arrow indicates the central part of the AS where thermocline depth is compared.




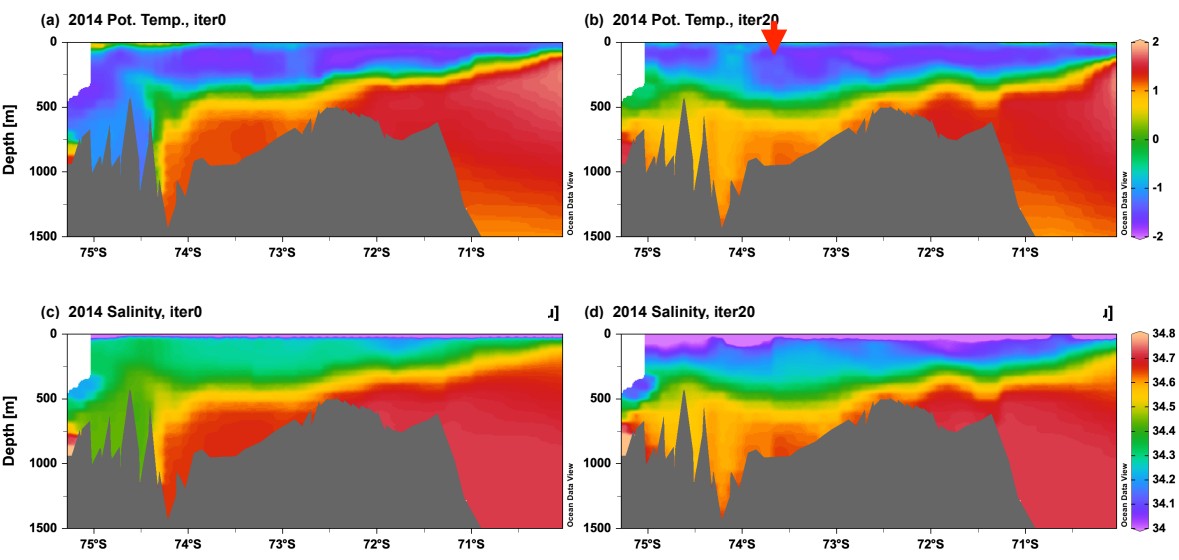

**Figure 5.** Simulated vertical sections of monthly mean potential temperature (top) and salinity (bottom) in January 2014 along the thick black line in Figure 1 for the (left) unoptimized and (right) iteration 20 simulations. The red arrow indicates the central part of the AS where thermocline depth is compared.





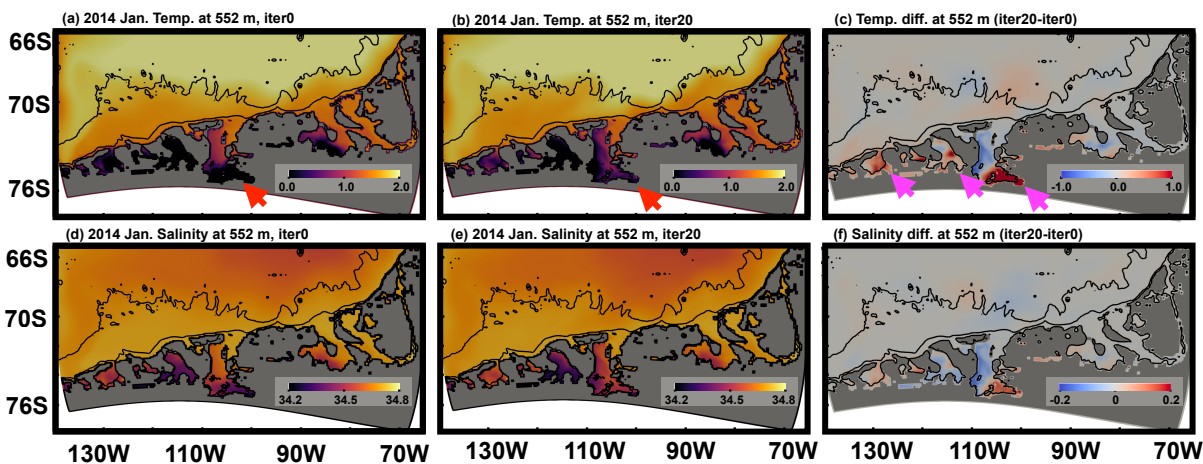

**Figure 6.** Simulated monthly mean (a,b) potential temperature and (d,e) salinity at 552 m depth in January 2014 for (left) unoptimized and (middle) iteration 20 simulations, respectively. (c) Potential temperature and (f) salinity differences between unoptimized and iteration 20 simulations. Bathymetric contours of 500, 2000, and 4000 m are shown in black. Red arrows indicate the PIIS front region and pink arrows indicate regions in the deep troughs in the AS. Potential temperature in these regions become warmer in iteration 20 simulation as mCDW intrusion into ice shelf cavities in the AS are correctly represented in the iteration 20 simulation.



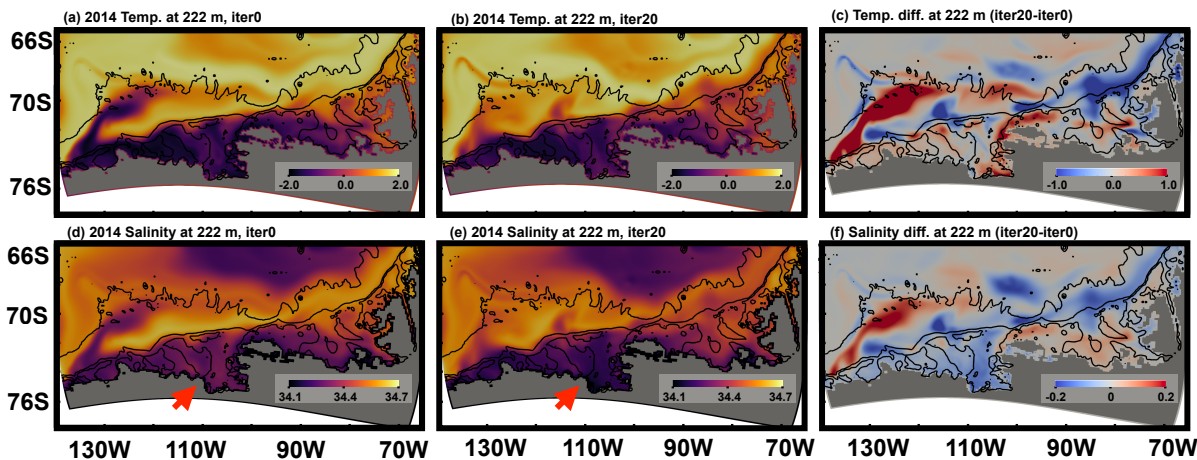

**Figure 7.** Simulated yearly mean (a,b) potential temperature and (d,e) salinity at 222 m depth in 2014 for (left) unoptimized and (middle) iteration 20 simulations, respectively. (c) Potential temperature and (f) salinity differences between unoptimized and iteration 20 simulations. Bathymetric contours of 500, 2000, and 4000 m are shown in black. Red arrows indicate the eastern AS region, where salinity becomes fresher by ∼0.1, showing an improvement of WW properties.





**Figure 8.** Simulated time series of (left) potential temperature and (right) salinity at BSR/iSTAR9 mooring locations for (a,b) iteration 0 (unoptimized), (c,d) iteration 10, and (g,f) iteration 20 simulations. Observed mooring timeseries are shown in Fig.12 and Fig.2c in Webber et al. (2017).





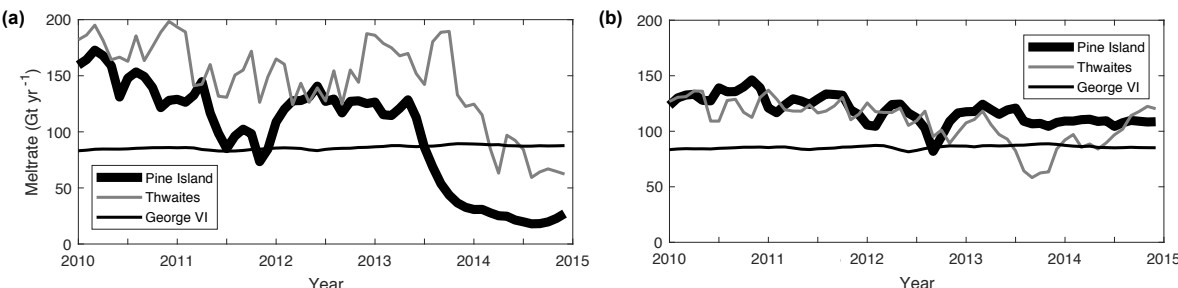

**Figure 9.** Simulated monthly mean basal melt rates from 2010 to 2015 for the Pine Island, Thwaites, and George Vl Ice Shelves for (a) unoptimized and (b) iteration 20 simulations.

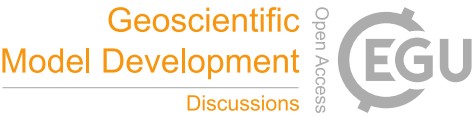

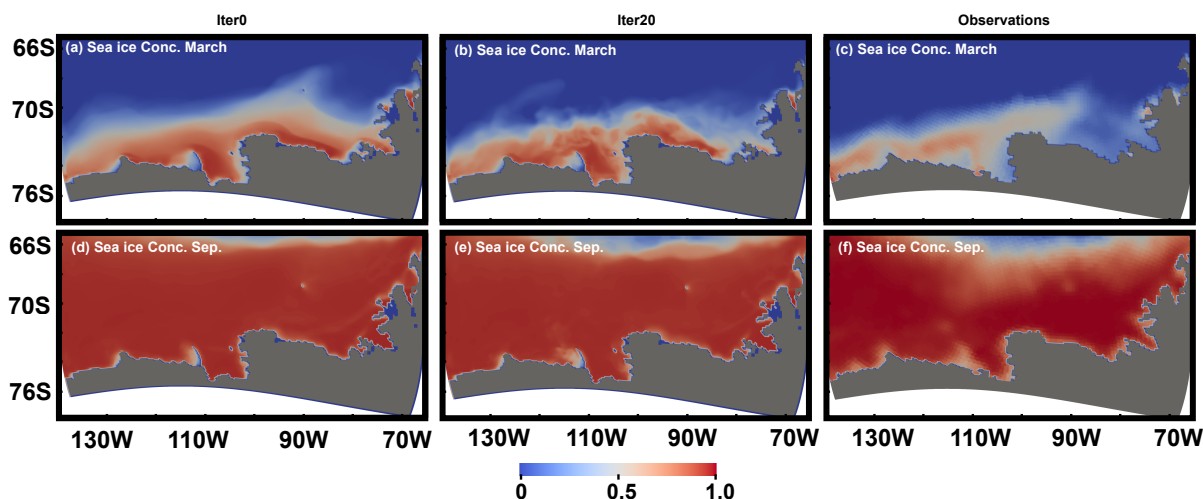

**Figure 10.** Simulated mean sea ice concentrations for (a,b) March and (d, e) September for unoptimized and iteration 20 simulations, respectively. The observed mean sea ice concentrations for (c) March and (f) September based on satellite sea ice concentration measurements between 2010–2014.



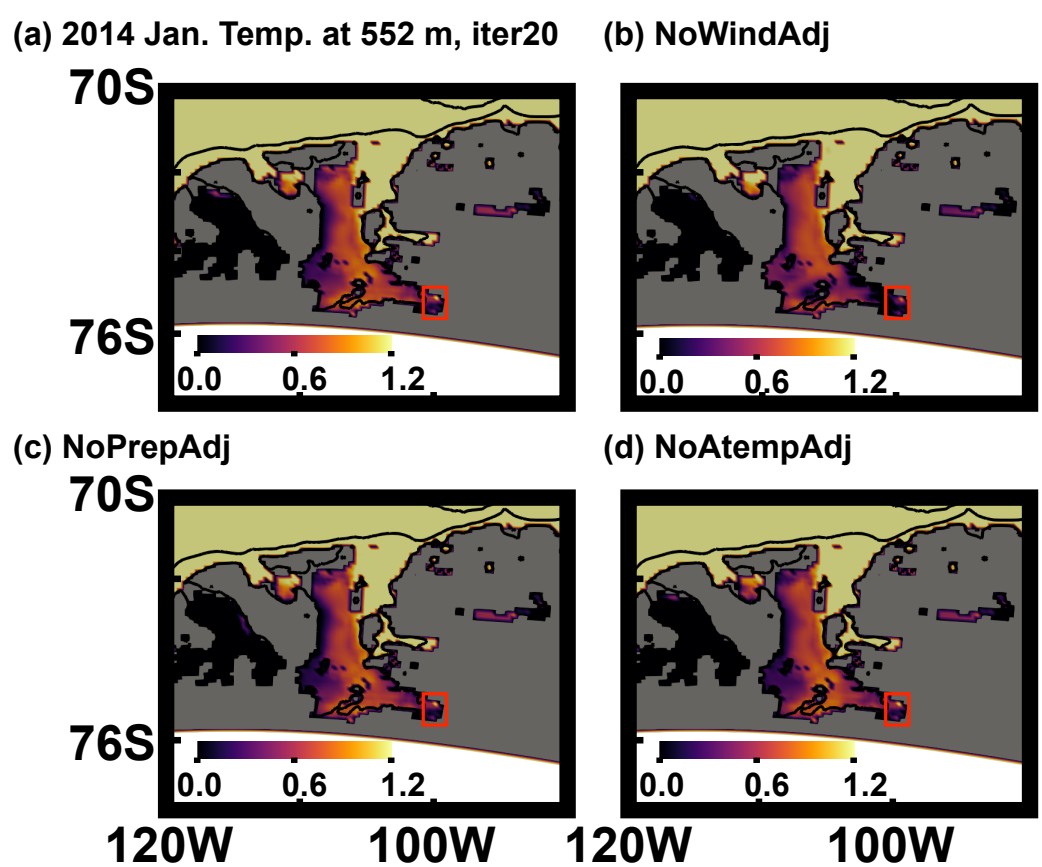

**Figure 11.** Simulated 2014 January mean 552 m potential temperature for (a) iteration 20, (b) NoWindAdj, (c) NoPrepAdj, and (d)NoAtempAdj simulations. Bathymetric contours of 500, 2000, and 4000 m are shown in black. Spatial averages of 552 m potential temperature are calculated for the region enclosed by the red boxes.



**Figure 12.** Time series of potential temperature at (a)477 m, (b)634 m, and (c) 909 m for unoptimized (iteration 0) and iteration 20 simulations. Time series of observed potential temperature at BSR5/iSTAR9 mooring sites at depths between (a) 400-500 m, (b) 600-700 m, and (c) 800-900 m are also shown in black. Time series of (d) spatially averaged (102.4-104.0°W, 74.8-75.0°S) sea ice concentration for unoptimized (iteration 0), iteration 20, and observations.





**Table 1.** Oceanographic datasets used for ocean state estimates.

| Measurements | Year | Reference | Locations |
|---|---|---|---|
| Ship-CTD | 2010 | Nakayama et al. (2013) | AS |
| Ship-CTD | 2010, 2011, 2012 | Dutrieux et al. (2014); Kim et al. (2017) | AS |
| Ship-CTD | 2014 | Heywood et al. (2016) | AS |
| Ship-CTD | 2010–2014 | e.g., Ducklow et al. (2012) | BS |
| Mooring | 2010–2014 | Webber et al. (2017) | AS |
| Mooring | 2012–2014 | Kim et al. (2017) | AS |
| Seal-CTD | 2014 | Mallett et al. (2018) | AS |
| Sea ice concentration | 2010–2014 | Cavalieri et al. (1996) | AS, BS |

**Table 2.** Model parameters used for ocean simulations. Most parameters are chosen based on Nakayama et al. (2017) and Zhang et al. (2018) with some adjustments.

| Parameter | |
|---|---|
| Horizontal diffusivity ( $m^2\ s^{-1}$ ) | 10 |
| Background horizontal viscosity ($m^2\ s^{-1}$) | 1000, 500, 100, 10 |
| Leith biharm non-dimensional viscosity factor | 0.0 |
| Modified Leith biharm non-dimensional viscosity factor | 0.0 |
| Background vertical diffusivity ($m^2\ s^{-1}$) | $5.456 \times 10^{-6}$ |
| Background vertical viscosity ($m^2\ s^{-1}$) | $1.0 \times 10^{-4}$ |
| KPP critical bulk Richardson Number | 0.3273 |
| KPP local Richardson Number limit for shear instability | 0.8358 |
| Bottom drag coefficient | $2.1 \times 10^{-3}$ |
| Ocean/air drag coefficient scaling factor | 0.508 |
| Air/sea ice drag coefficient | $1.0 \times 10^{-3}$ |
| Sea ice/ocean drag coefficient | $5.69 \times 10^{-3}$ |
| Sea ice salt concentration | 4.0 |
| Stanton number (stable) | 0.0492 |
| Stanton number (unstable) | 0.02506 |
| Dalton number | 0.0520 |
| Lead closing (m) | 1.24 |
| Ice strength (N $m^{-2}$ ) | $1.0 \times 10^{4}$ |
| Sea ice dry albedo | 0.84 |
| Sea ice wet albedo | 0.78 |
| Snow dry albedo | 0.90 |
| Snow wet albedo | 0.80 |




**Table 3.** Adjustments to model parameters in addition to optimization using adjoint sensitivities.

| Iterations | Adjustment |
| --- | --- |
| Iteration 10 | change background horizontal viscosity from 1000 m$^2$ s$^{-1}$ to 500 m$^2$ s$^{-1}$ |
| Iteration 11 | heat and salt transfer coefficients for PIIS and Thwaites ice shelf reduced by 70% and 63%, respectively |
| Iteration 15 | change background horizontal viscosity from 500 m$^2$ s$^{-1}$ to 100 m$^2$ s$^{-1}$ |
| Iteration 20 | change background horizontal viscosity from 100 m$^2$ s$^{-1}$ to 20 m$^2$ s$^{-1}$ |

**Table 4.** Satellite-based estimates of basal melt rate (Rignot et al., 2013) and model mean basal melt rates (2010–2014) for West Antarctic ice shelves for iteration 20. The values of heat transfer coefficient $\gamma_T$ used for the optimized simulation are also shown.

| Name | $\gamma_T$ (iterations 0–10) ($\times 10^{-4}$ m s$^{-1}$) | $\gamma_T$ (iterations 11–20) ($\times 10^{-4}$ m s$^{-1}$) | Observation based estimates (Rignot et al., 2013) (Gt yr$^{-1}$) | Optimized simulation (Gt yr$^{-1}$) |
| --- | --- | --- | --- | --- |
| George VI (Geo) | 0.11 | 0.11 | **89.0±17** | **85.6** |
| Wilkins (Wi) | 0.11 | 0.11 | **18.4±17** | **11.5** |
| Bach (Ba) | 0.57 | 0.57 | **10.4±1** | **11.8** |
| Stange (St) | 0.35 | 0.35 | **28.0±6** | **33.0** |
| Ferrigno (Fe) | 2.2 | 2.2 | **5.1±2** | **1.7** |
| Venable (Ve) | 0.35 | 0.35 | **19.4±2** | **20.0** |
| Abbot (Ab) | 0.27 | 0.27 | **51.8±19** | **53.0** |
| Cosgrove (Co) | 0.079 | 0.079 | **8.5±2** | **9.8** |
| Pine Island (PI) | 1.25 | 0.86 | **101.2±8** | **118.3** |
| Thwaites (Th) | 0.91 | 0.57 | **97.5±7** | **108.8** |
| Crosson (Cr) | 15.2 | 15.2 | **38.5±4** | **44.0** |
| Dotson (Do) | 3.3 | 3.3 | **45.2±4** | **40.6** |
| Getz (Get) | 0.26 | 0.26 | **144.9±14** | **128.1** |

**Table 5.** Description of all the sensitivity simulations.

| Case | |
| --- | --- |
| NoWindAdj | iteration 20 simulation but excluding adjustment for wind |
| NoPrepAdj | iteration 20 simulation but excluding adjustment for precipitation |
| NoAtempAdj | iteration 20 simulation but excluding adjustment for air temperature |