# Peer review of "Development of adjoint-based ocean state estimation for the Amundsen and Bellingshausen Seas and ice shelf cavities using MITgcm/ECCO (66j)"

_Geoscientific Model Development, 2020_

## Referee Comment (RC1)

In this work, the authors present a novel state estimate of the Amundsen and Bellingshausen Seas, along with the associated ice shelves and their cavities. Using techniques developed by the ECCO project, the authors iteratively optimize a numerical simulation of their target region using a suite of available observations. They show that these observational constraints increase the agreement between the numerical simulation and the observational suite, as evidenced by a decreasing cost function. The mean state of the model improves by this metric, although there is still room for improvement with respect to temporal variability on interannual and multi-year scales. Finally, the authors conduct a set of sensitivity experiments by turning off the optimization for selected aspects of the "control vector"; for example, they test the effect of optimizing wind by using the non-optimized wind, while leaving the other optimizations in place, and observing the resulting change in the cost function.

The paper addresses the highly relevant and challenging problem of simulating the Antarctic shelf seas and associated ice shelves. The Amundsen Sea region is especially relevant to the impacts of climate change and sea level rise, as it features especially high ice shelf melt rates; it is considered a vulnerable part of the ocean-ice system and is currently the focus of a UK-US Thwaites Glacier study, among other funded projects in the region. As such, the state estimate developed in this paper is highly relevant to a  pressing scientific problem. The work described here represents a significant step in ocean-ice modelling, as there have been very few prior attempts to incorporate ice shelves into state estimates. In this context, this modeling effort has been especially successful and is encouraging to see.

The methods and assumptions in the paper are clearly stated and appropriate for the process of state estimation. For example, changing the viscosities and melt rate coefficients for better agreement is a standard approach in adjoint modelling and state estimation. The results support the conclusions of the paper. An especially interesting result is that the adjustment of wind has the strongest impact on the ocean state, while the adjustment of heat flux has the strongest impact on the sea ice state.

The work is reproducible to an extent, as the authors have uploaded their work via Zenodo and the NASA data portal. The adjoint-driven optimization procedure uses an expensive commercial license to derive the adjoint code from the forward code; the cost of this license prohibits this work from being fully reproducible in terms of repeating the optimization process, but the open-source tools available for algorithmic differentiation of MITgcm are not yet competitive with the commercial tool. It is still standard practice in state estimation to use the commercial tool. That being said, the authors have taken the appropriate steps to make their state estimation work reproducible to the extent possible. One can run the non-optimized and the optimized simulations using the setup that they have provided. (I have not tried to run it myself, but from what I can see, all the required files have been uploaded.)

Overall, this is a highly relevant and exciting piece of work. The paper is mostly clear, but it would be an even stronger paper with some additional editing. I have made some specific suggestions below, along with a few questions about their state estimation procedure.

-- Specific questions, comments, and suggestions –

Abstract

(Lines 1-5) Change "is impacting" to "impacts". Add "the" before "global carbon".

(Line 20) Change "that that" to "that"

(Around line 25) Add "the" before "global carbon cycle". Add "the" before "ice shelf cavities".

(Line 42) Change "descriptions" to "representations"

(Line 43) I'm used to seeing this described as the "4DVAR method", as opposed to the "adjoint method". I think either one is fine, but its' good to be consistent and use the same term throughout the paper. Later in the paper this is called the "4DVAR method".

(Line 47) I've heard the suggestion that we should call this "algorithmic differentiation" instead of "automatic differentiation". Consider changing it if you agree.

(Between lines 60-65) Should "Green's function" be capitalized or not? Be consistent.

(Line 69) I'm not sure why the reference to Figure 2 is here. At this point, the concepts of iteration, cost functions, and the individual terms have not yet been introduced. It may be a bit confusing here. Add a quick definition of what you mean by "iterations" here. Otherwise people may get confused between numerical forward integration iterations and optimization iterations.

(Line 73) What is the advantage of using LLC270 here? Can you tell us more about this choice, please.

(Line 84) Why aren't these datasets used? Mentioning them raises the question as to why they weren't used.

(Line 92) What is a bipolar grid? Expand on this please.

(Lines 97-98) Please say more about why the ice barrier was necessary, for readers not familiar with the Nakayama study.

(Line 100) Sorry, what is the "ongoing LLC270 optimization"? Is this a global state estimate? What's the approximate resolution here?

(Line 101) Did you consider using the new ERA-5 product or perhaps MERRA? What was the reason for choosing ERA-Interim?

(Line 104) Here you call it the 4DVAR system for the first time. It's better to be consistent throughout the paper. Either call it the adjoint method or the 4DVAR method.

(Line 109) Did you use the M1QN3 that comes with MITgcm, or did you use Martin Losch's implementation of this approach?

(Line 111) Say more about this. What is the "pseudo-sea ice adjoint"?

(Line 117) How did you decide by how much to change the coefficients?

(Line 130) What is your point of comparison here? When you say "too cold and fresh", relative to what observations? Make it clear how you're doing this comparison.

(Lines 142-145) This is worded a bit confusingly. It's better to repeat the sentence instead of using the "respectively" construction. For example:

"For March in the AS, simulated sea ice in iteration 0 is larger than observations by X (Y% difference), and it iteration 20 it is larger by X (Y% difference). In the BS, ... (repeat)."

I would also include a percentage-wise comparison to give us a sense of the relative size of the difference.

(Line 150) Replace "2010-2014 simulation for unoptimized simulation" with "2010-2014 unoptimized simulation".

(Line 178) I would include these ice shelf melt rates as lines of reference on Figure 9. They could be straight horizontal lines, clearly labeled.

(Lines 190-205) It is a bit hard to tell from the text how the sensitivity experiments are done. Please be more explicit about how the tests were done and introduce the case names.

(224-226) So from what I understand, the sea ice cannot be adjusted directly in this state estimate, because there is only a pseudo-adjoint for the sea ice. Any changes in the simulated sea ice come from the adjustments of the surface forcing and ocean state. How do you think the solution would change if the sea ice could be included in the control vector? Since this is the discussion section, where perhaps a little informed speculation is allowed, can you speculate on how the addition of a sea ice adjoint would change the state estimate?

(Line 235) Change "is available" to "are available". Change "cost" to "the cost function" to be more explicit.

(Line 240) Add "the" in front of "seasonal".

(Line 242) The comparison with the mooring data in Figure 12 shows that the optimization procedure helped improve the agreement between the observations and the model state. I think it's worth emphasizing this positive improvement here.

(Line 246) You are using some moorings, correct? I assume this means that there are additional moorings that could be included in the future. Please could you clarify this?

-- Figures –

Figure 1. Add degree symbols to the latitudes and longitudes.

Figure 2 caption. Mention that the vertical scales are different in the different subplots. It's worth emphasizing.

Figure 3. This figure is not referenced in the paper, as far as I could tell. Add some text to the paper to describe this figure and to help the reader understand why you included it.

Figure 6 caption. Replace the last sentence with "In the iteration 20 simulation, potential temperature in these regions become warmer as mCDW intrusion into the ice shelf cavities in the AS are correctly represented."

Figure 8. This figure would be more impactful if you could plot the observed values in a subplot or two, instead of referring the reader to another figure. If this is possible, consider adding observational subplots for better comparison.

Figure 10. Add degree symbols to the lat/lon values. Also add a label to the colorbar; this will help readers quickly understand what they are looking at.

Figure 11. It is difficult to see the difference between these plots. Consider zooming in to the region where the changes are happening; at present there is a lot of blank space and space where the changes are very small. Please also consider plotting the differences instead; that may help clarify what the changes look like.

Figure 12. In panel (b), there are two black lines. It's not obvious to me why there are two. Please make them a bit different from each other and indicate where they come from.

-- Tables --

Table 4. Consider italicizing or otherwise visually indicating which basal melt rates that you changed, for quick visual reference. Or perhaps have these rows in bolt and remove the bold from the observational column and optimized column.

Very nice work!

---

## Author Comment (AC1)

**Response to the specific comments and corrections of the reviewers**
**(Italic: comment of reviewer, bold: our reply)**

**We greatly appreciate all very helpful and insightful comments by the reviewers.**

*Executive Editor (Comments to Author (shown to authors):*
*Dear authors,*

*in my role as Executive editor of GMD, I would like to bring to your attention our Editorial version 1.2:*

*https://www.geosci-model-dev.net/12/2215/2019/*

*This highlights some requirements of papers published in GMD, which is also available on the GMD website in the 'Manuscript Types' section:*

*http://www.geoscientific-model-development.net/submission/manuscript_types.html*

*In particular, please note that for your paper, the following requirement has not been met in the Discussions paper:*

- *"The main paper must give the model name and version number (or other unique identifier) in the title."*

*Please add a version numbers for MITgcm/ECCO in the title upon your revised submission to GMD.*

*Yours, Astrid Kerkweg*

**As suggested, we include version number in the title.**

---

## Author Comment (AC2)

**Response to the specific comments and corrections of the reviewers**

**(Italic: comment of reviewer, bold: our reply)**

**We greatly appreciate all very helpful and insightful comments by the reviewers.**

*Executive Editor (Comments to Author (shown to authors):*

*Dear authors,*

*in my role as Executive editor of GMD, I would like to bring to your attention our Editorial version 1.2:*

*https://www.geosci-model-dev.net/12/2215/2019/*

*This highlights some requirements of papers published in GMD, which is also available on the GMD website in the 'Manuscript Types' section:*

*http://www.geoscientific-model-development.net/submission/manuscript_types.html*

*In particular, please note that for your paper, the following requirement has not been met in the Discussions paper:*

- *"The main paper must give the model name and version number (or other unique identifier) in the title."*

*Please add a version numbers for MITgcm/ECCO in the title upon your revised submission to GMD.*

*Yours, Astrid Kerkweg*

**As suggested, we include version number in the title.**

*Reviewer #1 (Comments to Author (shown to authors):*

*In this work, the authors present a novel state estimate of the Amundsen and Bellingshausen Seas, along with the associated ice shelves and their cavities. Using techniques developed by the ECCO project, the authors iteratively optimize a numerical simulation of their target region using a suite of available observations. They show that these observational constraints increase the agreement between the numerical simulation and the observational suite, as evidenced by a decreasing cost function. The mean state of the model improves by this metric, although there is still room for improvement with respect to temporal variability on interannual and multi-year scales. Finally, the authors conduct a set of sensitivity experiments by turning off the optimization for selected aspects of the "control vector"; for example, they test the effect of optimizing wind by using the non-optimized wind, while leaving the other optimizations in place, and observing the resulting change in the cost function.*

*The paper addresses the highly relevant and challenging problem of simulating the Antarctic shelf seas and associated ice shelves. The Amundsen Sea region is especially relevant to the impacts of climate change and sea level rise, as it features especially high ice shelf melt rates; it is considered a vulnerable part of the ocean-ice system and is currently the focus of a UK-US Thwaites Glacier study, among other funded projects in the region. As such, the state estimate developed in this paper is highly relevant to a pressing scientific problem. The work described here represents a significant step in ocean-ice modelling, as there have been very few prior attempts to incorporate ice shelves into state estimates. In this context, this modeling effort has been especially successful and is encouraging to see.*

*The methods and assumptions in the paper are clearly stated and appropriate for the process of state estimation. For example, changing the viscosities and melt rate coefficients for better agreement is a standard approach in adjoint modelling and state estimation. The results support the conclusions of the paper. An especially interesting result is that the adjustment of wind has the strongest impact on the ocean state, while the adjustment of heat flux has the strongest impact on the sea ice state.*

*The work is reproducible to an extent, as the authors have uploaded their work via Zenodo and the NASA data portal. The adjoint-driven optimization procedure uses an expensive commercial license to derive the adjoint code from the forward code; the cost of this license prohibits this work from being fully reproducible in terms of repeating the optimization process, but the open-source tools available for algorithmic differentiation of MITgcm are not yet competitive with the commercial tool. It is still standard practice in state estimation to use the commercial tool. That being said, the authors have taken the appropriate steps to make their state estimation work reproducible to the extent possible. One can run the non-optimized and the optimized simulations using the setup that they have provided. (I have not tried to run it myself, but from what I can see, all the required files have been uploaded.)*

*Overall, this is a highly relevant and exciting piece of work. The paper is mostly clear, but it would be an even stronger paper with some additional editing. I have made some specific suggestions below, along with a few questions about their state estimation procedure.*

**Thank you very much for your encouraging comments.**

*-- Specific questions, comments, and suggestions –*
*Abstract*
*(Lines 1-5) Change "is impacting" to "impacts". Add "the" before "global carbon".*
**Done.**

*(Line 20) Change "that that" to "that"*
**Done.**

*(Around line 25) Add "the" before "global carbon cycle". Add "the" before "ice shelf cavities".*
**Done.**

*(Line 42) Change "descriptions" to "representations"*

**Done.**

*(Line 43) I'm used to seeing this described as the "4DVAR method", as opposed to the "adjoint method". I think either one is fine, but its' good to be consistent and use the same term throughout the paper. Later in the paper this is called the "4DVAR method".*
**We replaced 4D-VAR with adjoint but point out equivalence with 4D-Var method upon first use.**

*(Line 47) I've heard the suggestion that we should call this "algorithmic differentiation" instead of "automatic differentiation". Consider changing it if you agree.*
**We added "aka algorithmic differentiation". The two terms are somewhat equivalent but "automatic differentiation" appears to be more widely used for source-to-source transformation tools.**

*(Between lines 60-65) Should "Green's function" be capitalized or not? Be consistent.*
**Done. We now use "Green's functions" throughout.**

*(Line 69) I'm not sure why the reference to Figure 2 is here. At this point, the concepts of iteration, cost functions, and the individual terms have not yet been introduced. It may be a bit confusing here. Add a quick definition of what you mean by "iterations" here. Otherwise people may get confused between numerical forward integration iterations and optimization iterations.*
**We agree and have removed the reference to Figure 2 on line 69 in the revised manuscript.**

*(Line 73) What is the advantage of using LLC270 here? Can you tell us more about this choice, please.*
**In the revised manuscript, we added: "Using the ECCO LLC270 solution both provides lateral boundary conditions for this study as well as enabling this work to be a stepping stone towards improved representation of ice-shelf-ocean interactions in ECCO global-ocean retrospective analyses."**

*(Line 84) Why aren't these datasets used? Mentioning them raises the question as to why they weren't used.*

**Authors were not aware of these datasets when we started this project. We have modified sentence to make this clearer yet let readers be aware that other datasets are becoming available for inclusion in future studies.**

*(Line 92) What is a bipolar grid? Expand on this please.*

**Bipolar refers to the way the ocean south of 70S is represented in global ECCO LLC model configurations --- see Fig. 2, right panel in https://doi.org/10.5194/gmd-8-3071-2015. The most important information here is that "in the AS and BS domain, horizontal grid spacing is approximately 10 km". Thus we remove the mention of bipolar grid in the revised manuscript.**

*(Lines 97-98) Please say more about why the ice barrier was necessary, for readers not familiar with the Nakayama study.*

**Done.**

*(Line 100) Sorry, what is the "ongoing LLC270 optimization"? Is this a global state estimate? What's the approximate resolution here?*

**We now introduce ECCO LLC270 earlier in the manuscript and mention that it is a global-ocean state estimate with 1/3-deg horizontal grid spacing. Reference repeated here for clarity.**

*(Line 101) Did you consider using the new ERA-5 product or perhaps MERRA? What was the reason for choosing ERA-Interim?*

**ERA-Interim was first-guess for the ECCO LLC270 optimization (Zhang et al., 2018). ERA-5 was not yet available when the ECCO LLC270 optimization was initiated. More recent ECCO ocean state estimates have started using MERRA-2 and ERA-5. But note that the ECCO machinery adjusts atmospheric state, presumably reducing dependence on first-guess atmospheric state.**

*(Line 104) Here you call it the 4DVAR system for the first time. It's better to be consistent throughout the paper. Either call it the adjoint method or the 4DVAR method.*
**We replace 4D-VAR with adjoint.**

*(Line 109) Did you use the M1QN3 that comes with MITgcm, or did you use Martin Losch's implementation of this approach?*
**Yes. As mentioned in the manuscript, we use "tangent linear model backward in time (Le Dimet and Talagrand, 1986) and is used with the quasi-Newton M1QN3 conjugate- gradient algorithm".**

*(Line 111) Say more about this. What is the "pseudo-sea ice adjoint"? (Line 117) How did you decide by how much to change the coefficients?*
**We now include a detailed explanation of pseudo sea ice.**

*(Line 130) What is your point of comparison here? When you say "too cold and fresh", relative to what observations? Make it clear how you're doing this comparison.*
**Done.**

*(Lines 142-145) This is worded a bit confusingly. It's better to repeat the sentence instead of using the "respectively" construction. For example:*

*"For March in the AS, simulated sea ice in iteration 0 is larger than observations by X (Y% difference), and it iteration 20 it is larger by X (Y% difference). In the BS, ... (repeat)."*

*I would also include a percentage-wise comparison to give us a sense of the relative size of the difference.*
**Done.**

*(Line 150) Replace "2010-2014 simulation for unoptimized simulation" with "2010-2014 unoptimized simulation".*
**Done.**

*(Line 178) I would include these ice shelf melt rates as lines of reference on Figure 9. They could be straight horizontal lines, clearly labeled.*

**Since these melt rates are estimated based on snapshot oceanographic observations with assumptions and we think this may include large source of error. We would like to keep Figure 9 as it is.**

*(Lines 190-205) It is a bit hard to tell from the text how the sensitivity experiments are done. Please be more explicit about how the tests were done and introduce the case names.*

**Done (See Lines 209-211).**

*(224-226) So from what I understand, the sea ice cannot be adjusted directly in this state estimate, because there is only a pseudo-adjoint for the sea ice. Any changes in the simulated sea ice come from the adjustments of the surface forcing and ocean state. How do you think the solution would change if the sea ice could be included in the control vector? Since this is the discussion section, where perhaps a little informed speculation is allowed, can you speculate on how the addition of a sea ice adjoint would change the state estimate?*

**Done (See Lines 253-255).**

*(Line 235) Change "is available" to "are available". Change "cost" to "the cost function" to be more explicit.*

**Done.**

*(Line 240) Add "the" in front of "seasonal".*

**Done.**

*(Line 242) The comparison with the mooring data in Figure 12 shows that the optimization procedure helped improve the agreement between the observations and the model state. I think it's worth emphasizing this positive improvement here.*

**We include the reference to Figure 12 to emphasize this point (Line 268).**

*(Line 246) You are using some moorings, correct? I assume this means that there are additional moorings that could be included in the future. Please could you clarify this?*
**Done (Line 275).**

*-- Figures –*
*Figure 1. Add degree symbols to the latitudes and longitudes.*
**Done.**

*Figure 2 caption. Mention that the vertical scales are different in the different subplots. It's worth emphasizing.*
**Done.**

*Figure 3. This figure is not referenced in the paper, as far as I could tell. Add some text to the paper to describe this figure and to help the reader understand why you included it.*
**This figure is referenced in the revised manuscript (see Lines 90, 146, and 170).**

*Figure 6 caption. Replace the last sentence with "In the iteration 20 simulation, potential temperature in these regions become warmer as mCDW intrusion into the ice shelf cavities in the AS are correctly represented."*
**Done.**

*Figure 8. This figure would be more impactful if you could plot the observed values in a subplot or two, instead of referring the reader to another figure. If this is possible, consider adding observational subplots for better comparison.*
**Adding observational subplots requires extensive work including complex interpolation because two mooring observations are conducted off the PIIS with many instruments at various depths with different observational periods, frequencies, and termination (machine failures), etc. In the current version of our manuscript, we only discuss model-data agreement in terms of mean state. We do not think it is necessary to add a separate figure at this stage. It is likely needed for our future work when we are able to achieve better model-data**

**agreement in terms of interannual variability of thermocline depths and mCDW properties.**

*Figure 10. Add degree symbols to the lat/lon values. Also add a label to the colorbar; this will help readers quickly understand what they are looking at.*
**Done.**

*Figure 11. It is difficult to see the difference between these plots. Consider zooming in to the region where the changes are happening; at present there is a lot of blank space and space where the changes are very small. Please also consider plotting the differences instead; that may help clarify what the changes look like.*
**Done.**

*Figure 12. In panel (b), there are two black lines. It's not obvious to me why there are two. Please make them a bit different from each other and indicate where they come from.*
**Measurements from multiple sensors are plotted in panels a, b, and c located at different depths. In the revised manuscript, we indicate observed depths in the figure captions and make clear why there are multiple black lines in each panel.**

*-- Tables --*

*Table 4. Consider italicizing or otherwise visually indicating which basal melt rates that you changed, for quick visual reference. Or perhaps have these rows in bolt and remove the bold from the observational column and optimized column.*
**Done.**

*Very nice work!*

---

## Author Comment (AC3)

**Response to the specific comments and corrections of the reviewers**
**(Italic: comment of reviewer, bold: our reply)**

**We greatly appreciate all very helpful and insightful comments by the reviewers.**

*Executive Editor (Comments to Author (shown to authors):*
*Dear authors,*

*in my role as Executive editor of GMD, I would like to bring to your attention our Editorial version 1.2:*

*https://www.geosci-model-dev.net/12/2215/2019/*

*This highlights some requirements of papers published in GMD, which is also available on the GMD website in the 'Manuscript Types' section:*

*http://www.geoscientific-model-development.net/submission/manuscript_types.html*

*In particular, please note that for your paper, the following requirement has not been met in the Discussions paper:*

- *"The main paper must give the model name and version number (or other unique identifier) in the title."*

*Please add a version numbers for MITgcm/ECCO in the title upon your revised submission to GMD.*

*Yours, Astrid Kerkweg*

**As suggested, we include version number in the title.**

*Reviewer #2 (Comments to Author (shown to authors):*

*This manuscript presents the results from the adjoint-based data assimilation/ocean state estimation for the Amundsen and the Bellingshausen Sea where ice-shelf-ocean interactions are important. The strength of this work lies in that it is the first to attempt the data assimilation in the regions since the similar effort using Green's functions, by overcoming the limitation of this low-dimensional estimation approach. The manuscript is generally well-written and represents a substantial contribution to the modeling communities. However, my first major concern is that despite the importance of serving as the first adjoint-based state estimation effort in these regions, the authors did not go into much detail on model skill assessment, which appears rather descriptive by simply comparing the model outputs to observations. Other major comments are concerned with the lack of the discussion of optimized parameters and sensitivity experiments to varying model initial conditions/parameter guesses.*

*1. Model skill assessment*

*Model skill is discussed with regard to the model-observations misfits and the model improvement capturing key oceanographic characteristics by the optimized simulation compared to the initial (unoptimized) simulation. However, much of this discussion appear descriptive, and I would suggest that the authors calculate a series of skill metrics for a more quantitative model skill assessment. For example, what is the formula of the model cost function, and how were the components (e.g., means, weights, standard deviations of the assimilated data types) calculated within? I would recommend that the authors examine the univariate model metrics (e.g., r, RMSD, the reliability index, the average error or bias, the average absolute error, and the modeling efficiency; in Stow et al. 2009, J. Mar. Sys. 76:4-15) and add the Taylor diagrams showing r, RMSD, and the normalized standard deviation, before and after optimization (one set based on the unoptimized simulation and the other based on the final optimized simulation).*

**We agree that in the previous version of the manuscript, it was not clear how the cost function is defined. We revised the manuscript to state that we use a cost**

**function following Forget et al., 2015. The weights for potential-temperature and salinity observations are prescribed as a function of depth and are estimated based on the standard deviation of the simulated properties in the model domain (Lines 107-111). Following suggestions from the reviewer, we calculated r, RMSD, and normalized standard deviation before and after optimization. We also include a Taylor diagram and add discussion (Lines 243-249).**

*2. Discussion of optimized parameters*
*It is nice to see the in-depth discussion of possible causes of model-observation mismatch, the limitation of the adjoint-optimization methodology, and the utilization of limited observational data. As the authors mentioned in the manuscript, I see that the important contribution of this work to the broad modeling community is to provide a better set of model parameter estimates for the AS and BS regions in future global ECCO optimizations. I would suggest that the authors report the optimized parameter values with uncertainty ranges (assuming those are calculated from the inverse Hessian matrix/approximation) and whether these values make sense scientifically. Also, the current modeled fields do not have any uncertainties, perhaps because only 1 optimization experiment was performed. If the uncertainties of the optimized parameters can be calculated, how the random perturbations within the range of the optimized parameters can impact the modeled fields?*

**One drawback of the adjoint method is that model uncertainty cannot be calculated directly when obtaining optimized parameters. One component of uncertainty can, in theory, be calculated by obtaining the second derivative of the cost function but this would involve an unrealistic amount of computations. For ECCO-v4 (which has a grid cell count similar to that of our higher-resolution regional domain), the dimension of the state vector at each time step is greater than N=11 million. Updating the state and its covariance would require running the model N+1 times at each time step, as described in Wunsch (2018).  More practical and less formal ways of obtaining uncertainty for ocean state estimates**

**are discussed in Wunsch (2018) but their application is beyond the scope of the current manuscript.**

**Carl Wunsch (2018) Towards determining uncertainties in global oceanic mean values of heat, salt, and surface elevation, Tellus A: Dynamic Meteorology and Oceanography, 70:1, 1-14, DOI: 10.1080/16000870.2018.1471911**

*3. Sensitivity tests to varying initial conditions/parameter guesses*
*It is great to see that the authors added several sensitivity trials to test the relative importance of air temperature, precipitation, and winds in better matching the region-specific, nonlinear processes. However, I wonder if starting in another place in parameter space would lead to a significantly different local minimum of the cost function. I understand that the time and effort of conducting even 1 optimization experiment (20 iterations for this study) can be significant, so I would not suggest that the authors do a large number of new optimization from different initial conditions/parameter guesses, but still would like to see a reasonable number of trials to ensure the robustness of the optimized model solution presented in the study.*

**We agree that this is a very good suggestion but it would require a substantial amount of additional work — the adjoint-model-based ocean state estimate presented herein was achieved after ~5 years of development and computations. Instead we revise the manuscript in Lines 276-277 to suggest this as future work.**

*Other comments:*

*Figures 4-5 do not show observations but consistently referenced in the sentences discussing the model-observation misfits. I would suggest that the authors include the detailed characteristics and patterns of the observational data as searate figures or sentences rather than referencing figures from other papers.*
**We do not include observations as these sections are obtained from different locations and can not be compared as done in Figures 4 and 5. We instead include a few sentences to indicate that detailed model-data comparisons were**

**conducted in Nakayama et al., 2017 (Lines 136 and 143-144).**

*Line 142: I am not convinced that the figure shows much better agreements to the sea-ice observations in the optimized simulation. For September, how much of overestimation at iteration 0? Closer to observation after optimization by how much? Please be more quantitative.*
**We revised the manuscript as suggested (Lines 157-158).**

*Line 168: The fact that heat and salt transfer coefficients changed at iteration 11 conflicts with line 166. Please rephrase this section. The authors discussed the reason why whose parameters had to change in lines 116-118 in the Methods section, but these all should consistently appear together in the methods, or in the results.*
**We revised the manuscript as suggested (Line 183).**

*Line 190-197: Instead of providing the absolute cost function values (not very meaningful without the full presentation of the model-observation misfit calculations) please calculate the percent reduction in the cost function. Also, I am not sure why Table 5 is needed. The summary of what changed as part of sensitivity experiments can be directly stated in the discussion, and maybe with the cost function reduction reported in Table 5.*
**We revised the manuscript as suggested. We now include percent increase in Table 5, as these sensitivity experiments tend to increase the cost compared to the control (CTRL) experiment.**

*Figure 12: indicate the depth in a-c.*
**Done. Depths are indicated in the figure caption.**

---

## Author Comment (AC4)

Response to the specific comments and corrections of the reviewers (Italic: comment of reviewer, bold: our reply)

**We greatly appreciate all very helpful and insightful comments by the reviewers.**

Executive Editor (Comments to Author (shown to authors): Dear authors,

in my role as Executive editor of GMD, I would like to bring to your attention our Editorial version 1.2:

https://www.geosci-model-dev.net/12/2215/2019/

This highlights some requirements of papers published in GMD, which is also available on the GMD website in the 'Manuscript Types' section:

http://www.geoscientific-model-development.net/submission/manuscript\_types.html

In particular, please note that for your paper, the following requirement has not been met in the Discussions paper:

• "The main paper must give the model name and version number (or other unique identifier) in the title."

Please add a version numbers for MITgcm/ECCO in the title upon your revised submission to GMD.

Yours, Astrid Kerkweg

As suggested, we include version number in the title.

*Reviewer* #1 (Comments to Author (shown to authors):

In this work, the authors present a novel state estimate of the Amundsen and Bellingshausen Seas, along with the associated ice shelves and their cavities. Using techniques developed by the ECCO project, the authors iteratively optimize a numerical simulation of their target region using a suite of available observations. They show that these observational constraints increase the agreement between the numerical simulation and the observational suite, as evidenced by a decreasing cost function. The mean state of the model improves by this metric, although there is still room for improvement with respect to temporal variability on interannual and multi-year scales. Finally, the authors conduct a set of sensitivity experiments by turning off the optimization for selected aspects of the "control vector"; for example, they test the effect of optimizing wind by using the non-optimized wind, while leaving the other optimizations in place, and observing the resulting change in the cost function.

The paper addresses the highly relevant and challenging problem of simulating the Antarctic shelf seas and associated ice shelves. The Amundsen Sea region is especially relevant to the impacts of climate change and sea level rise, as it features especially high ice shelf melt rates; it is considered a vulnerable part of the ocean-ice system and is currently the focus of a UK-US Thwaites Glacier study, among other funded projects in the region. As such, the state estimate developed in this paper is highly relevant to a pressing scientific problem. The work described here represents a significant step in ocean-ice modelling, as there have been very few prior attempts to incorporate ice shelves into state estimates. In this context, this modeling effort has been especially successful and is encouraging to see.

The methods and assumptions in the paper are clearly stated and appropriate for the process of state estimation. For example, changing the viscosities and melt rate coefficients for better agreement is a standard approach in adjoint modelling and state estimation. The results support the conclusions of the paper. An especially interesting result is that the adjustment of wind has the strongest impact on the ocean state, while the adjustment of heat flux has the strongest impact on the sea ice state.

The work is reproducible to an extent, as the authors have uploaded their work via Zenodo and the NASA data portal. The adjoint-driven optimization procedure uses an expensive commercial license to derive the adjoint code from the forward code; the cost of this license prohibits this work from being fully reproducible in terms of repeating the optimization process, but the open-source tools available for algorithmic differentiation of MITgcm are not yet competitive with the commercial tool. It is still standard practice in state estimation to use the commercial tool. That being said, the authors have taken the appropriate steps to make their state estimation work reproducible to the extent possible. One can run the non-optimized and the optimized simulations using the setup that they have provided. (I have not tried to run it myself, but from what I can see, all the required files have been uploaded.)

Overall, this is a highly relevant and exciting piece of work. The paper is mostly clear, but it would be an even stronger paper with some additional editing. I have made some specific suggestions below, along with a few questions about their state estimation procedure.

**Thank you very much for your encouraging comments.**

-- Specific questions, comments, and suggestions – Abstract (Lines 1-5) Change "is impacting" to "impacts". Add "the" before "global carbon". **Done.**

(Line 20) Change "that that" to "that" **Done.**

(Around line 25) Add "the" before "global carbon cycle". Add "the" before "ice shelf cavities".

**Done.**

(Line 42) Change "descriptions" to "representations"

**Done.**

(Line 43) I'm used to seeing this described as the "4DVAR method", as opposed to the "adjoint method". I think either one is fine, but its' good to be consistent and use the same term throughout the paper. Later in the paper this is called the "4DVAR method". We replaced 4D-VAR with adjoint but point out equivalence with 4D-Var method upon first use.

(Line 47) I've heard the suggestion that we should call this "algorithmic differentiation" instead of "automatic differentiation". Consider changing it if you agree.
We added "aka algorithmic differentiation". The two terms are somewhat equivalent but "automatic differentiation" appears to be more widely used for source-to-source transformation tools.

(Between lines 60-65) Should "Green's function" be capitalized or not? Be consistent. **Done. We now use "Green's functions" throughout.**

(Line 69) I'm not sure why the reference to Figure 2 is here. At this point, the concepts of iteration, cost functions, and the individual terms have not yet been introduced. It may be a bit confusing here. Add a quick definition of what you mean by "iterations" here. Otherwise people may get confused between numerical forward integration iterations and optimization iterations.

We agree and have removed the reference to Figure 2 on line 69 in the revised manuscript.

(Line 73) What is the advantage of using LLC270 here? Can you tell us more about this choice, please.

In the revised manuscript, we added: "Using the ECCO LLC270 solution both provides lateral boundary conditions for this study as well as enabling this work to be a stepping stone towards improved representation of ice-shelf-ocean interactions in ECCO global-ocean retrospective analyses." (Line 84) Why aren't these datasets used? Mentioning them raises the question as to why they weren't used.

Authors were not aware of these datasets when we started this project. We have modified sentence to make this clearer yet let readers be aware that other datasets are becoming available for inclusion in future studies.

**(Line 92) What is a bipolar grid? Expand on this please.**

Bipolar refers to the way the ocean south of 70S is represented in global ECCO LLC model configurations --- see Fig. 2, right panel in https://doi.org/10.5194/gmd- 8-3071-2015. The most important information here is that "in the AS and BS domain, horizontal grid spacing is approximately 10 km". Thus we remove the mention of bipolar grid in the revised manuscript.

(Lines 97-98) Please say more about why the ice barrier was necessary, for readers not familiar with the Nakayama study. **Done.**

(Line 100) Sorry, what is the "ongoing LLC270 optimization"? Is this a global state estimate? What's the approximate resolution here?

We now introduce ECCO LLC270 earlier in the manuscript and mention that it is a global-ocean state estimate with 1/3-deg horizontal grid spacing. Reference repeated here for clarity.

(Line 101) Did you consider using the new ERA-5 product or perhaps MERRA? What was the reason for choosing ERA-Interim?

ERA-Interim was first-guess for the ECCO LLC270 optimization (Zhang et al., 2018). ERA-5 was not yet available when the ECCO LLC270 optimization was initiated. More recent ECCO ocean state estimates have started using MERRA-2 and ERA-5. But note that the ECCO machinery adjusts atmospheric state, presumably reducing dependence on first-guess atmospheric state.

(Line 104) Here you call it the 4DVAR system for the first time. It's better to be consistent throughout the paper. Either call it the adjoint method or the 4DVAR method. **We replace 4D-VAR with adjoint.**

(Line 109) Did you use the M1QN3 that comes with MITgcm, or did you use Martin Losch's implementation of this approach?

Yes. As mentioned in the manuscript, we use "tangent linear model backward in time (Le Dimet and Talagrand, 1986) and is used with the quasi-Newton M1QN3 conjugate- gradient algorithm".

(Line 111) Say more about this. What is the "pseudo-sea ice adjoint"? (Line 117) How did you decide by how much to change the coefficients?

We now include a detailed explanation of pseudo sea ice in Lines 119-124.

(Line 130) What is your point of comparison here? When you say "too cold and fresh", relative to what observations? Make it clear how you're doing this comparison. **Done.**

(Lines 142-145) This is worded a bit confusingly. It's better to repeat the sentence instead of using the "respectively" construction. For example:

"For March in the AS, simulated sea ice in iteration 0 is larger than observations by X (Y% difference), and it iteration 20 it is larger by X (Y% difference). In the BS, ... (repeat)."

I would also include a percentage-wise comparison to give us a sense of the relative size of the difference.

**Done.**

(Line 150) Replace "2010-2014 simulation for unoptimized simulation" with "2010-2014 unoptimized simulation".

**Done.**

(Line 178) I would include these ice shelf melt rates as lines of reference on Figure 9. They could be straight horizontal lines, clearly labeled.

Since these melt rates are estimated based on snapshot oceanographic observations with assumptions and we think this may include large source of error. We would like to keep Figure 9 as it is.

(Lines 190-205) It is a bit hard to tell from the text how the sensitivity experiments are done. Please be more explicit about how the tests were done and introduce the case names.

**Done (See Lines 211-213).**

(224-226) So from what I understand, the sea ice cannot be adjusted directly in this state estimate, because there is only a pseudo-adjoint for the sea ice. Any changes in the simulated sea ice come from the adjustments of the surface forcing and ocean state. How do you think the solution would change if the sea ice could be included in the control vector? Since this is the discussion section, where perhaps a little informed speculation is allowed, can you speculate on how the addition of a sea ice adjoint would change the state estimate?

Done (See Lines 255-257).

(Line 235) Change "is available" to "are available". Change "cost" to "the cost function" to be more explicit.

**Done.**

(Line 240) Add "the" in front of "seasonal". **Done.**

(Line 242) The comparison with the mooring data in Figure 12 shows that the optimization procedure helped improve the agreement between the observations and the model state. I think it's worth emphasizing this positive improvement here. We include the reference to Figure 12 to emphasize this point (Line 270). (Line 246) You are using some moorings, correct? I assume this means that there are additional moorings that could be included in the future. Please could you clarify this? **Done (Line 277).**

-- Figures –

*Figure 1. Add degree symbols to the latitudes and longitudes.* **Done.**

Figure 2 caption. Mention that the vertical scales are different in the different subplots. It's worth emphasizing.

**Done.**

Figure 3. This figure is not referenced in the paper, as far as I could tell. Add some text to the paper to describe this figure and to help the reader understand why you included it.

This figure is referenced in the revised manuscript (see Lines 90, 148, and 172).

Figure 6 caption. Replace the last sentence with "In the iteration 20 simulation, potential temperature in these regions become warmer as mCDW intrusion into the ice shelf cavities in the AS are correctly represented."

**Done.**

Figure 8. This figure would be more impactful if you could plot the observed values in a subplot or two, instead of referring the reader to another figure. If this is possible, consider adding observational subplots for better comparison.

Adding observational subplots requires extensive work including complex interpolation because two mooring observations are conducted off the PIIS with many instruments at various depths with different observational periods, frequencies, and termination (machine failures), etc. In the current version of our manuscript, we only discuss model-data agreement in terms of mean state. We do not think it is necessary to add a separate figure at this stage. It is likely needed for our future work when we are able to achieve better model-data agreement in terms of interannual variability of thermocline depths and mCDW properties.

Figure 10. Add degree symbols to the lat/lon values. Also add a label to the colorbar; this will help readers quickly understand what they are looking at. **Done.**

Figure 11. It is difficult to see the difference between these plots. Consider zooming in to the region where the changes are happening; at present there is a lot of blank space and space where the changes are very small. Please also consider plotting the differences instead; that may help clarify what the changes look like. **Done.**

Figure 12. In panel (b), there are two black lines. It's not obvious to me why there are two. Please make them a bit different from each other and indicate where they come from.

Measurements from multiple sensors are plotted in panels a, b, and c located at different depths. In the revised manuscript, we indicate observed depths in the figure captions and make clear why there are multiple black lines in each panel.

-- Tables --

Table 4. Consider italicizing or otherwise visually indicating which basal melt rates that you changed, for quick visual reference. Or perhaps have these rows in bolt and remove the bold from the observational column and optimized column. **Done.**

Very nice work!

Reviewer #2 (Comments to Author (shown to authors):

This manuscript presents the results from the adjoint-based data assimilation/ocean state estimation for the Amundsen and the Bellingshausen Sea where ice-shelf-ocean interactions are important. The strength of this work lies in that it is the first to attempt the data assimilation in the regions since the similar effort using Green's functions, by overcoming the limitation of this low-dimensional estimation approach. The manuscript is generally well-written and represents a substantial contribution to the modeling communities. However, my first major concern is that despite the importance of serving as the first adjoint-based state estimation effort in these regions, the authors did not go into much detail on model skill assessment, which appears rather descriptive by simply comparing the model outputs to observations. Other major comments are concerned with the lack of the discussion of optimized parameters and sensitivity experiments to varying model initial conditions/parameter guesses.

**1. Model skill assessment**

Model skill is discussed with regard to the model-observations misfits and the model improvement capturing key oceanographic characteristics by the optimized simulation compared to the initial (unoptimized) simulation. However, much of this discussion appear descriptive, and I would suggest that the authors calculate a series of skill metrics for a more quantitative model skill assessment. For example, what is the formula of the model cost function, and how were the components (e.g., means, weights, standard deviations of the assimilated data types) calculated within? I would recommend that the authors examine the univariate model metrics (e.g., r, RMSD, the reliability index, the average error or bias, the average absolute error, and the modeling efficiency; in Stow et al. 2009, J. Mar. Sys. 76:4-15) and add the Taylor diagrams showing r, RMSD, and the normalized standard deviation, before and after optimization (one set based on the unoptimized simulation and the other based on the final optimized simulation).

We agree that in the previous version of the manuscript, it was not clear how the cost function is defined. We revised the manuscript to state that we use a cost

function following Forget et al., 2015. The weights for potential-temperature and salinity observations are prescribed as a function of depth and are estimated based on the standard deviation of the simulated properties in the model domain (Lines 107-116). Following suggestions from the reviewer, we calculated r, RMSD, and normalized standard deviation before and after optimization. We also include a Taylor diagram and add discussion (Lines 245-251).

**2. Discussion of optimized parameters**

It is nice to see the in-depth discussion of possible causes of model-observation mismatch, the limitation of the adjoint-optimization methodology, and the utilization of limited observational data. As the authors mentioned in the manuscript, I see that the important contribution of this work to the broad modeling community is to provide a better set of model parameter estimates for the AS and BS regions in future global ECCO optimizations. I would suggest that the authors report the optimized parameter values with uncertainty ranges (assuming those are calculated from the inverse Hessian matrix/approximation) and whether these values make sense scientifically. Also, the current modeled fields do not have any uncertainties, perhaps because only 1 optimization experiment was performed. If the uncertainties of the optimized parameters can be calculated, how the random perturbations within the range of the optimized parameters can impact the modeled fields?

A partial measure of uncertainty for the modeled fields is now provided with the improved and more quantitative comparison of the optimized solution with observations. A comprehensive estimate of parametric uncertainty is (1) unfortunately too expensive to carry out and (2) would probably not be very meaningful given paucity of observations and strong dependence of posterior uncertainty on prior error estimates.

(1) Although parametric uncertainty can in theory be calculated by obtaining the second derivative of the cost function (i.e., based on the Hessian matrix as the reviewer points out), this would involve an unrealistic amount of computations. For ECCO-v4 (which has a grid cell count similar to that of our higher-resolution regional domain), the dimension of the state vector at each time step is greater than N=11 million. Updating the state and its covariance would require running the model N+1 times at each time step, as described in Wunsch (2018). Similarly a Monte-Carlo approach for estimating parametric uncertainty (e.g., by carrying out multiple optimizations with various prior assumptions as the reviewer suggests) is also not practical computationally.

(2) Assuming that parametric uncertainty could be obtained, the second difficulty is the paucity of observations, which make meaningful prior and posterior uncertainty estimates very uncertain. As discussed in Menemenlis and Chechelnitsky (2000), a major obstacle to obtaining statistically significant uncertainty is the large uncertainty of sample covariance matrices,  $O(2\sigma^2/p)$ where  $\sigma^2$  is the variance and p is the degrees of freedom. In other words, the number of statistically significant uncertainty parameters that can be estimated is orders of magnitude smaller than the total number of independent observations.

Until more observations or more computational cycles are available, the best we can do is the more quantitative model-data comparison, which has been added in the revised manuscript.

Carl Wunsch (2018). Towards determining uncertainties in global oceanic mean values of heat, salt, and surface elevation. *Tellus A: Dynamic Meteorology and Oceanography*, 70:1, 1-14, DOI: 10.1080/16000870.2018.1471911

D. Menemenlis and M. Chechelnitsky (2000). Error Estimates for an Ocean General Circulation Model from Altimeter and Acoustic Tomography Data. *Monthly Weather Review*, 128:3, 763–778. DOI: 10.1175/1520-0493(2000)128<0763:EEFAOG>2.0.CO;2

3. Sensitivity tests to varying initial conditions/parameter guesses It is great to see that the authors added several sensitivity trials to test the relative importance of air temperature, precipitation, and winds in better matching the regionspecific, nonlinear processes. However, I wonder if starting in another place in parameter space would lead to a significantly different local minimum of the cost function. I understand that the time and effort of conducting even 1 optimization experiment (20 iterations for this study) can be significant, so I would not suggest that the authors do a large number of new optimization from different initial conditions/parameter guesses, but still would like to see a reasonable number of trials to ensure the robustness of the optimized model solution presented in the study.

We agree that this is a very good suggestion but it would require a substantial amount of additional work — the adjoint-model-based ocean state estimate presented herein was achieved after ~5 years of development and computations. Instead we revise the manuscript in Lines 278-279 to suggest this as future work.

Other comments:

Figures 4-5 do not show observations but consistently referenced in the sentences discussing the model-observation misfits. I would suggest that the authors include the detailed characteristics and patterns of the observational data as searate figures or sentences rather than referencing figures from other papers.

We do not include observations as these sections are obtained from different locations and can not be compared as done in Figures 4 and 5. We instead include a few sentences to indicate that detailed model-data comparisons were conducted in Nakayama et al., 2017 (Lines 138 and 145-146).

Line 142: I am not convinced that the figure shows much better agreements to the seaice observations in the optimized simulation. For September, how much of overestimation at iteration 0? Closer to observation after optimization by how much? Please be more quantitative.

We revised the manuscript as suggested (Lines 158-160).

Line 168: The fact that heat and salt transfer coefficients changed at iteration 11 conflicts with line 166. Please rephrase this section. The authors discussed the reason why whose parameters had to change in lines 116-118 in the Methods section, but these all should consistently appear together in the methods, or in the results. **We revised the manuscript as suggested (Line 185).**

Line 190-197: Instead of providing the absolute cost function values (not very meaningful without the full presentation of the model-observation misfit calculations) please calculate the percent reduction in the cost function. Also, I am not sure why Table 5 is needed. The summary of what changed as part of sensitivity experiments can be directly stated in the discussion, and maybe with the cost function reduction reported in Table 5.

We revised the manuscript as suggested. We now include percent increase in Table 5, as these sensitivity experiments tend to increase the cost compared to the control (CTRL) experiment.

Figure 12: indicate the depth in a-c.

Done. Depths are indicated in the figure caption.